# Adversarial Training and Robustness for Multiple Perturbations

**Florian Tramèr**
Stanford University

**Dan Boneh**
Stanford University

## Abstract

Defenses against adversarial examples, such as adversarial training, are typically tailored to a single perturbation type (e.g., small $\ell_\infty$-noise). For other perturbations, these defenses offer no guarantees and, at times, even increase the model's vulnerability. Our aim is to understand the reasons underlying this robustness trade-off, and to train models that are simultaneously robust to multiple perturbation types.

We prove that a trade-off in robustness to different types of $\ell_p$-bounded and spatial perturbations must exist in a natural and simple statistical setting. We corroborate our formal analysis by demonstrating similar robustness trade-offs on MNIST and CIFAR10. We propose new multi-perturbation adversarial training schemes, as well as an efficient attack for the $\ell_1$-norm, and use these to show that models trained against multiple attacks fail to achieve robustness competitive with that of models trained on each attack individually. In particular, we find that adversarial training with first-order $\ell_\infty, \ell_1$ and $\ell_2$ attacks on MNIST achieves merely 50% robust accuracy, partly because of gradient-masking. Finally, we propose *affine attacks* that linearly interpolate between perturbation types and further degrade the accuracy of adversarially trained models.

## 1 Introduction

Adversarial examples [37, 15] are proving to be an inherent blind-spot in machine learning (ML) models. Adversarial examples highlight the tendency of ML models to learn superficial and brittle data statistics [19, 13, 18], and present a security risk for models deployed in cyber-physical systems (e.g., virtual assistants [5], malware detectors [16] or ad-blockers [39]).

Known successful defenses are tailored to a specific perturbation type (e.g., a small $\ell_p$-ball [25, 28, 42] or small spatial transforms [11]). These defenses provide empirical (or certifiable) robustness guarantees for one perturbation type, but typically offer no guarantees against other attacks [35, 31]. Worse, increasing robustness to one perturbation type has sometimes been found to increase vulnerability to others [11, 31]. This leads us to the central problem considered in this paper:

> *Can we achieve adversarial robustness to different types of perturbations simultaneously?*

Note that even though prior work has attained robustness to different perturbation types [25, 31, 11], these results may not compose. For instance, an ensemble of two classifiers—each of which is robust to a single type of perturbation—may be robust to neither perturbation. Our aim is to study the extent to which it is possible to learn models that are *simultaneously* robust to multiple types of perturbation.

To gain intuition about this problem, we first study a simple and natural classification task, that has been used to analyze trade-offs between standard and adversarial accuracy [41], and the sample-complexity of adversarial generalization [30]. We define *Mutually Exclusive Perturbations (MEPs)* as pairs of perturbation types for which robustness to one type implies vulnerability to the other. For this task, we prove that $\ell_\infty$ and $\ell_1$-perturbations are MEPs and that $\ell_\infty$-perturbations and input rotations

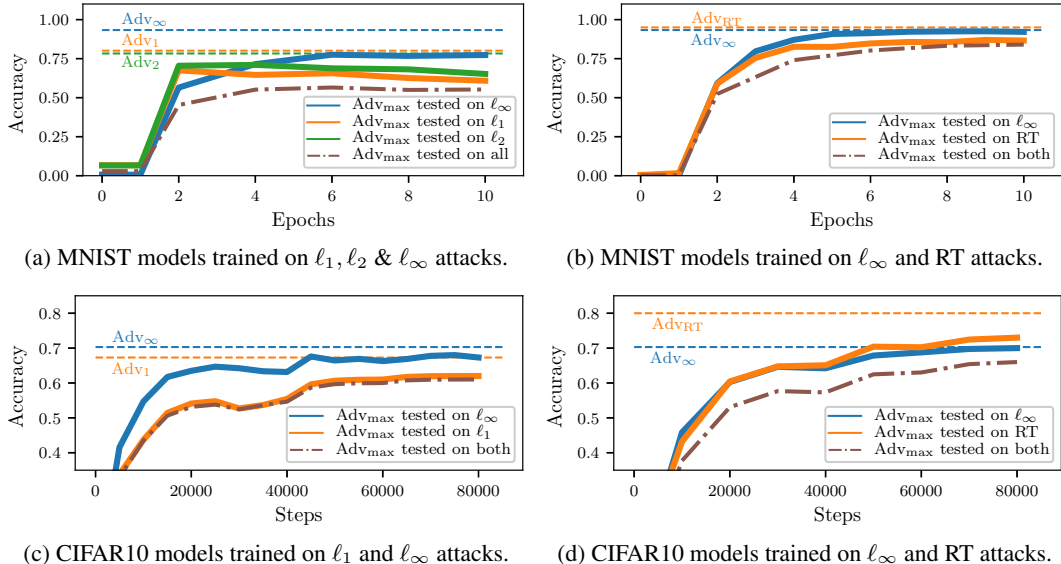

(a) MNIST models trained on $\ell_1, \ell_2$ & $\ell_\infty$ attacks.

(b) MNIST models trained on $\ell_\infty$ and RT attacks.

(c) CIFAR10 models trained on $\ell_1$ and $\ell_\infty$ attacks.

(d) CIFAR10 models trained on $\ell_\infty$ and RT attacks.

Figure 1: **Robustness trade-off on MNIST (top) and CIFAR10 (bottom).** For a union of $\ell_p$-balls (left), or of $\ell_\infty$-noise and rotation-translations (RT) (right), we train models $\text{Adv}_{\max}$ on the strongest perturbation-type for each input. We report the test accuracy of $\text{Adv}_{\max}$ against each individual perturbation type (solid line) and against their union (dotted brown line). The vertical lines show the adversarial accuracy of models trained and evaluated on a single perturbation type.

and translations [11] are also MEPs. Moreover, for these MEP pairs, we find that robustness to either perturbation type requires fundamentally different features. The existence of such a trade-off for this simple classification task suggests that it may be prevalent in more complex statistical settings.

To complement our formal analysis, we introduce new adversarial training schemes for multiple perturbations. For each training point, these schemes build adversarial examples for all perturbation types and then train either on all examples (the "avg" strategy) or only the worst example (the "max" strategy). These two strategies respectively minimize the *average* error rate across perturbation types, or the error rate against an adversary that picks the worst perturbation type for each input.

For adversarial training to be practical, we also need efficient and strong attacks [25]. We show that Projected Gradient Descent [22, 25] is inefficient in the $\ell_1$-case, and design a new attack, *Sparse $\ell_1$ Descent* (SLIDE), that is both efficient and competitive with strong optimization attacks [8],

We experiment with MNIST and CIFAR10. MNIST is an interesting case-study, as *distinct* models from prior work attain strong robustness to all perturbations we consider [25, 31, 11], yet no *single* classifier is robust to all attacks [31, 32, 11]. For models trained on multiple $\ell_p$-attacks ($\ell_1, \ell_2, \ell_\infty$ for MNIST, and $\ell_1, \ell_\infty$ for CIFAR10), or on both $\ell_\infty$ and spatial transforms [11], we confirm a noticeable robustness trade-off. Figure 1 plots the test accuracy of models $\text{Adv}_{\max}$ trained using our "max" strategy. In all cases, robustness to multiple perturbations comes at a cost—usually of 5-10% additional error—compared to models trained against each attack individually (the horizontal lines).

Robustness to $\ell_1, \ell_2$ and $\ell_\infty$-noise on MNIST is a striking failure case, where the robustness trade-off is compounded by *gradient-masking* [27, 40, 1]. Extending prior observations [25, 31, 23], we show that models trained against an $\ell_\infty$-adversary learn representations that *mask gradients* for attacks in other $\ell_p$-norms. When trained against first-order $\ell_1, \ell_2$ and $\ell_\infty$-attacks, the model learns to resist $\ell_\infty$-attacks while giving the illusion of robustness to $\ell_1$ and $\ell_2$ attacks. This model only achieves 52% accuracy when evaluated on gradient-free attacks [3, 31]. This shows that, unlike previously thought [41], adversarial training with strong first-order attacks can suffer from gradient-masking. We thus argue that attaining robustness to $\ell_p$-noise on MNIST requires new techniques (e.g., training on expensive gradient-free attacks, or scaling certified defenses to multiple perturbations).

MNIST has sometimes been said to be a poor dataset for evaluating adversarial examples defenses, as some attacks are easy to defend against (e.g., input-thresholding or binarization works well for $\ell_\infty$-attacks [41, 31]). Our results paint a more nuanced view: the simplicity of these $\ell_\infty$-defenses

becomes a disadvantage when training against multiple $\ell_p$-norms. We thus believe that MNIST should not be abandoned as a benchmark just yet. Our inability to achieve multi-$\ell_p$ robustness for this simple dataset raises questions about the viability of scaling current defenses to more complex tasks.

Looking beyond adversaries that choose from a union of perturbation types, we introduce a new *affine adversary* that may linearly interpolate between perturbations (e.g., by compounding $\ell_\infty$-noise with a small rotation). We prove that for locally-linear models, robustness to a union of $\ell_p$-perturbations implies robustness to affine attacks. In contrast, affine combinations of $\ell_\infty$ and spatial perturbations are provably stronger than either perturbation individually. We show that this discrepancy translates to neural networks trained on real data. Thus, in some cases, attaining robustness to a union of perturbation types remains insufficient against a more creative adversary that composes perturbations.

Our results show that despite recent successes in achieving robustness to single perturbation types, many obstacles remain towards attaining truly robust models. Beyond the robustness trade-off, efficient computational scaling of current defenses to multiple perturbations remains an open problem.

The code used for all of our experiments can be found here: `https://github.com/ftramer/MultiRobustness`

Proofs of all theorems, experimental setups, and additional experiments are in the full version of this extended abstract [38].

# 2 Theoretical Limits to Multi-perturbation Robustness

We study statistical properties of adversarial robustness in a natural statistical model introduced in [41], and which exhibits many phenomena observed on real data, such as trade-offs between robustness and accuracy [41] or a higher sample complexity for robust generalization [31]. This model also proves useful in analyzing and understanding adversarial robustness for multiple perturbations. Indeed, we prove a number of results that correspond to phenomena we observe on real data, in particular trade-offs in robustness to different $\ell_p$ or rotation-translation attacks [11].

We follow a line of works that study distributions for which adversarial examples exist *unconditionally* [41, 21, 33, 12, 14, 26]. These distributions, including ours, are much simpler than real-world data, and thus need not be evidence that adversarial examples are inevitable in practice. Rather, we hypothesize that current ML models are highly vulnerable to adversarial examples because they learn superficial data statistics [19, 13, 18] that share some properties of these simple distributions.

In prior work, a robustness trade-off for $\ell_\infty$ and $\ell_2$-noise is shown in [21] for data distributed over two concentric spheres. Our conceptually simpler model has the advantage of yielding results beyond $\ell_p$-norms (e.g., for spatial attacks) and which apply symmetrically to both classes. Building on work by Xu et al. [43], Demontis et al. [9] show a robustness trade-off for dual norms (e.g., $\ell_\infty$ and $\ell_1$-noise) in linear classifiers.

## 2.1 Adversarial Risk for Multiple Perturbation Models

Consider a classification task for a distribution $\mathcal{D}$ over examples $\boldsymbol{x} \in \mathbb{R}^d$ and labels $y \in [C]$. Let $f : \mathbb{R}^d \to [C]$ denote a classifier and let $l(f(\boldsymbol{x}), y)$ be the zero-one loss (i.e., $\mathbb{1}_{f(\boldsymbol{x}) \neq y}$).

We assume $n$ *perturbation types*, each characterized by a set $S$ of allowed perturbations for an input $\boldsymbol{x}$. The set $S$ can be an $\ell_p$-ball [37, 15] or capture other perceptually small transforms such as image rotations and translations [11]. For a perturbation $\boldsymbol{r} \in S$, an adversarial example is $\hat{\boldsymbol{x}} = \boldsymbol{x} + \boldsymbol{r}$ (this is pixel-wise addition for $\ell_p$ perturbations, but can be a more complex operation, e.g., for rotations).

For a perturbation set $S$ and model $f$, we define $\mathcal{R}_{\text{adv}}(f; S) := \mathbb{E}_{(\boldsymbol{x},y) \sim \mathcal{D}} \left[ \max_{\boldsymbol{r} \in S} l(f(\boldsymbol{x} + \boldsymbol{r}), y) \right]$ as the adversarial error rate. To extend $\mathcal{R}_{\text{adv}}$ to multiple perturbation sets $S_1, \ldots, S_n$, we can consider the *average* error rate for each $S_i$, denoted $\mathcal{R}_{\text{adv}}^{\text{avg}}$. This metric most clearly captures the trade-off in robustness across independent perturbation types, but is not the most appropriate from a security perspective on adversarial examples. A more natural metric, denoted $\mathcal{R}_{\text{adv}}^{\text{max}}$, is the error rate against an adversary that picks, for each input, the worst perturbation from the *union* of the $S_i$. More formally,

$$\mathcal{R}_{\text{adv}}^{\text{max}}(f; S_1, \ldots, S_n) := \mathcal{R}_{\text{adv}}(f; \cup_i S_i) , \quad \mathcal{R}_{\text{adv}}^{\text{avg}}(f; S_1, \ldots, S_n) := \frac{1}{n} \sum_i \mathcal{R}_{\text{adv}}(f; S_i) . \quad (1)$$

Most results in this section are *lower bounds* on $\mathcal{R}_{\text{adv}}^{\text{avg}}$, which also hold for $\mathcal{R}_{\text{max}}^{\text{avg}}$ since $\mathcal{R}_{\text{adv}}^{\text{max}} \geq \mathcal{R}_{\text{adv}}^{\text{avg}}$.

Two perturbation types $S_1, S_2$ are *Mutually Exclusive Perturbations (MEPs)*, if $\mathcal{R}_{\text{adv}}^{\text{avg}}(f; S_1, S_2) \geq 1/|C|$ for all models $f$ (i.e., no model has non-trivial average risk against both perturbations).

## 2.2 A binary classification task

We analyze the adversarial robustness trade-off for different perturbation types in a natural statistical model introduced by Tsipras et al. [41]. Their binary classification task consists of input-label pairs $(\boldsymbol{x}, y)$ sampled from a distribution $\mathcal{D}$ as follows (note that $\mathcal{D}$ is $(d+1)$-dimensional):

$$y \overset{u.a.r}{\sim} \{-1, +1\}, \quad x_0 = \begin{cases} +y, & \text{w.p. } p_0, \\ -y, & \text{w.p. } 1 - p_0 \end{cases}, \quad x_1, \ldots, x_d \overset{i.i.d}{\sim} \mathcal{N}(y\eta, 1) , \tag{2}$$

where $p_0 \geq 0.5$, $\mathcal{N}(\mu, \sigma^2)$ is the normal distribution and $\eta = \alpha/\sqrt{d}$ for some positive constant $\alpha$.

For this distribution, Tsipras et al. [41] show a trade-off between standard and adversarial accuracy (for $\ell_\infty$ attacks), by drawing a distinction between the "robust" feature $x_0$ that small $\ell_\infty$-noise cannot manipulate, and the "non-robust" features $x_1, \ldots, x_d$ that can be fully overridden by small $\ell_\infty$-noise.

## 2.3 Small $\ell_\infty$ and $\ell_1$ Perturbations are Mutually Exclusive

The starting point of our analysis is the observation that the robustness of a feature depends on the considered perturbation type. To illustrate, we recall two classifiers from [41] that operate on disjoint feature sets. The first, $f(\boldsymbol{x}) = \text{sign}(x_0)$, achieves accuracy $p_0$ for all $\ell_\infty$-perturbations with $\epsilon < 1$ but is highly vulnerable to $\ell_1$-perturbations of size $\epsilon \geq 1$. The second classifier, $h(\boldsymbol{x}) = \text{sign}(\sum_{i=1}^{d} x_i)$ is robust to $\ell_1$-perturbations of average norm below $\mathbb{E}[\sum_{i=1}^{d} x_i] = \Theta(\sqrt{d})$, yet it is fully subverted by a $\ell_\infty$-perturbation that shifts the features $x_1, \ldots, x_d$ by $\pm 2\eta = \Theta(1/\sqrt{d})$. We prove that this tension between $\ell_\infty$ and $\ell_1$ robustness, and of the choice of "robust" features, is inherent for this task:

**Theorem 1.** *Let $f$ be a classifier for $\mathcal{D}$. Let $S_\infty$ be the set of $\ell_\infty$-bounded perturbations with $\epsilon = 2\eta$, and $S_1$ the set of $\ell_1$-bounded perturbations with $\epsilon = 2$. Then, $\mathcal{R}_{adv}^{avg}(f; S_\infty, S_1) \geq 1/2$ .*

The proof is in Appendix F. The bound shows that no classifier can attain better $\mathcal{R}_{\text{adv}}^{\text{avg}}$ (and thus $\mathcal{R}_{\text{adv}}^{\text{max}}$) than a trivial constant classifier $f(x) = 1$, which satisfies $\mathcal{R}_{\text{adv}}(f; S_\infty) = \mathcal{R}_{\text{adv}}(f; S_1) = 1/2$.

Similar to [9], our analysis extends to arbitrary dual norms $\ell_p$ and $\ell_q$ with $1/p + 1/q = 1$ and $p < 2$. The perturbation required to flip the features $x_1, \ldots, x_n$ has an $\ell_p$ norm of $\Theta(d^{\frac{1}{p} - \frac{1}{2}}) = \omega(1)$ and an $\ell_q$ norm of $\Theta(d^{\frac{1}{q} - \frac{1}{2}}) = \Theta(d^{\frac{1}{2} - \frac{1}{p}}) = o(1)$. Thus, feature $x_0$ is more robust than features $x_1, \ldots, x_n$ with respect to the $\ell_q$-norm, whereas for the dual $\ell_p$-norm the situation is reversed.

## 2.4 Small $\ell_\infty$ and Spatial Perturbations are (nearly) Mutually Exclusive

We now analyze two other orthogonal perturbation types, $\ell_\infty$-noise and rotation-translations [11]. In some cases, increasing robustness to $\ell_\infty$-noise has been shown to decrease robustness to rotation-translations [11]. We prove that such a trade-off is inherent for our binary classification task.

To reason about rotation-translations, we assume that the features $x_i$ form a 2D grid. We also let $x_0$ be distributed as $\mathcal{N}(y, \alpha^{-2})$, a technicality that does not qualitatively change our prior results. Note that the distribution of the features $x_1, \ldots, x_d$ is permutation-invariant. Thus, the only power of a rotation-translation adversary is to "move" feature $x_0$. Without loss of generality, we identify a small rotation-translation of an input $\boldsymbol{x}$ with a permutation of its features that sends $x_0$ to one of $N$ fixed positions (e.g., with translations of $\pm 3$px as in [11], $x_0$ can be moved to $N = 49$ different positions).

A model can be robust to these permutations by ignoring the $N$ positions that feature $x_0$ can be moved to, and focusing on the remaining permutation-invariant features. Yet, this model is vulnerable to $\ell_\infty$-noise, as it ignores $x_0$. In turn, a model that relies on feature $x_0$ can be robust to $\ell_\infty$-perturbations, but is vulnerable to a spatial perturbation that "hides" $x_0$ among other features. Formally, we show:

**Theorem 2.** *Let $f$ be a classifier for $\mathcal{D}$ (with $x_0 \sim \mathcal{N}(y, \alpha^{-2})$). Let $S_\infty$ be the set of $\ell_\infty$-bounded perturbations with $\epsilon = 2\eta$, and $S_{RT}$ be the set of perturbations for an RT adversary with budget $N$. Then, $\mathcal{R}_{adv}^{avg}(f; S_\infty, S_{RT}) \geq 1/2 - O(1/\sqrt{N})$ .*

The proof, given in Appendix G, is non-trivial and yields an asymptotic lower-bound on $\mathcal{R}_{\text{adv}}^{\text{avg}}$. We can also provide tight numerical estimates for concrete parameter settings (see Appendix G.1).

## 2.5 Affine Combinations of Perturbations

We defined $\mathcal{R}_{adv}^{max}$ as the error rate against an adversary that may choose a different perturbation type for each input. If a model were robust to this adversary, what can we say about the robustness to a more creative adversary that *combines* different perturbation types? To answer this question, we introduce a new adversary that mixes different attacks by linearly interpolating between perturbations.

For a perturbation set $S$ and $\beta \in [0,1]$, we denote $\beta \cdot S$ the set of perturbations scaled down by $\beta$. For an $\ell_p$-ball with radius $\epsilon$, this is the ball with radius $\beta \cdot \epsilon$. For rotation-translations, the attack budget $N$ is scaled to $\beta \cdot N$. For two sets $S_1, S_2$, we define $S_{affine}(S_1, S_2)$ as the set of perturbations that compound a perturbation $\boldsymbol{r}_1 \in \beta \cdot S_1$ with a perturbation $\boldsymbol{r}_2 \in (1 - \beta) \cdot S_2$, for any $\beta \in [0,1]$.

Consider one adversary that chooses, for each input, $\ell_p$ or $\ell_q$-noise from balls $S_p$ and $S_q$, for $p, q > 0$. The affine adversary picks perturbations from the set $S_{affine}$ defined as above. We show:

**Claim 3.** *For a linear classifier $f(\boldsymbol{x}) = sign(\boldsymbol{w}^T \boldsymbol{x} + b)$, we have $\mathcal{R}_{adv}^{max}(f; S_p, S_q) = \mathcal{R}_{adv}(f; S_{affine})$.*

Thus, for linear classifiers, robustness to a union of $\ell_p$-perturbations implies robustness to affine adversaries (this holds for any distribution). The proof, in Appendix H extends to models that are *locally linear* within balls $S_p$ and $S_q$ around the data points. For the distribution $\mathcal{D}$ of Section 2.2, we can further show that there are settings (distinct from the one in Theorem 1) where: (1) robustness against a union of $\ell_\infty$ and $\ell_1$-perturbations is possible; (2) this requires the model to be non-linear; (3) yet, robustness to affine adversaries is impossible (see Appendix I for details). Our experiments in Section 4 show that neural networks trained on CIFAR10 have a behavior that is consistent with locally-linear models, in that they are as robust to affine adversaries as against a union of $\ell_p$-attacks.

In contrast, compounding $\ell_\infty$ and spatial perturbations yields a stronger attack, even for linear models:

**Theorem 4.** *Let $f(\boldsymbol{x}) = sign(\boldsymbol{w}^T \boldsymbol{x} + b)$ be a linear classifier for $\mathcal{D}$ (with $x_0 \sim \mathcal{N}(y, \alpha^{-2})$). Let $S_\infty$ be some $\ell_\infty$-ball and $S_{RT}$ be rotation-translations with budget $N > 2$. Define $S_{affine}$ as above. Assume $w_0 > w_i > 0, \forall i \in [1, d]$. Then $\mathcal{R}_{adv}(f; S_{affine}) > \mathcal{R}_{adv}^{max}(f; S_\infty, S_{RT})$.*

This result (the proof is in Appendix J) draws a distinction between the strength of affine combinations of $\ell_p$-noise, and combinations of $\ell_\infty$ and spatial perturbations. It also shows that robustness to a union of perturbations can be insufficient against a more creative affine adversary. These results are consistent with behavior we observe in models trained on real data (see Section 4).

## 3 New Attacks and Adversarial Training Schemes

We complement our theoretical results with empirical evaluations of the robustness trade-off on MNIST and CIFAR10. To this end, we first introduce new adversarial training schemes tailored to the multi-perturbation risks defined in Equation (1), as well as a novel attack for the $\ell_1$-norm.

**Multi-perturbation adversarial training.** Let

$$\hat{\mathcal{R}}_{adv}(f; S) = \sum_{i=1}^{m} \max_{\boldsymbol{r} \in S} L(f(\boldsymbol{x}^{(i)} + \boldsymbol{r}), y^{(i)}) \, ,$$

bet the empirical adversarial risk, where $L$ is the training loss and $D$ is the training set. For a single perturbation type, $\hat{\mathcal{R}}_{adv}$ can be minimized with *adversarial training* [25]: the maximal loss is approximated by an attack procedure $\mathcal{A}(\boldsymbol{x})$, such that $\max_{\boldsymbol{r} \in S} L(f(\boldsymbol{x} + \boldsymbol{r}), y) \approx L(f(\mathcal{A}(\boldsymbol{x})), y)$.

For $i \in [1, d]$, let $\mathcal{A}_i$ be an attack for the perturbation set $S_i$. The two multi-attack robustness metrics introduced in Equation (1) immediately yield the following natural adversarial training strategies:

1. **"Max" strategy:** For each input $\boldsymbol{x}$, we train on the strongest adversarial example from all attacks, i.e., the max in $\hat{\mathcal{R}}_{adv}$ is replaced by $L(f(\mathcal{A}_{k^*}(\boldsymbol{x})), y)$, for $k^* = \arg\max_k L(f(\mathcal{A}_k(\boldsymbol{x})), y)$.
2. **"Avg" strategy:** This strategy simultaneously trains on adversarial examples from all attacks. That is, the max in $\hat{\mathcal{R}}_{adv}$ is replaced by $\frac{1}{n} \sum_{i=1}^{n} L(f(\mathcal{A}_i(\boldsymbol{x}), y))$.

**The sparse $\ell_1$-descent attack (SLIDE).** Adversarial training is contingent on a *strong* and *efficient* attack. Training on weak attacks gives no robustness [40], while strong optimization attacks (e.g., [6,

**Input**: Input $\boldsymbol{x} \in [0,1]^d$, steps $k$, step-size $\gamma$, percentile $q$, $\ell_1$-bound $\epsilon$
**Output**: $\hat{\boldsymbol{x}} = \boldsymbol{x} + \boldsymbol{r}$ s.t. $\|\boldsymbol{r}\|_1 \leq \epsilon$

---

$\boldsymbol{r} \leftarrow \boldsymbol{0}^d$
**for** $1 \leq i \leq k$ **do**
    $\boldsymbol{g} \leftarrow \nabla_{\boldsymbol{r}} L(\theta, \boldsymbol{x} + \boldsymbol{r}, y)$
    $e_i = \text{sign}(g_i)$ if $|g_i| \geq P_q(|\boldsymbol{g}|)$, else $0$
    $\boldsymbol{r} \leftarrow \boldsymbol{r} + \gamma \cdot \boldsymbol{e}/\|\boldsymbol{e}\|_1$
    $\boldsymbol{r} \leftarrow \Pi_{S_1^\epsilon}(\boldsymbol{r})$
**end**

---

**Algorithm 1: The Sparse $\ell_1$ Descent Attack (SLIDE).** $P_q(|\boldsymbol{g}|)$ denotes the $q^{\text{th}}$ percentile of $|\boldsymbol{g}|$ and $\Pi_{S_1^\epsilon}$ is the projection onto the $\ell_1$-ball (see [10]).

8]) are prohibitively expensive. Projected Gradient Descent (PGD) [22, 25] is a popular choice of attack that is both efficient and produces strong perturbations. To complement our formal results, we want to train models on $\ell_1$-perturbations. Yet, we show that the $\ell_1$-version of PGD is highly inefficient, and propose a better approach suitable for adversarial training.

PGD is a *steepest descent* algorithm [24]. In each iteration, the perturbation is updated in the steepest descent direction $\arg\max_{\|\boldsymbol{v}\|\leq 1} \boldsymbol{v}^T \boldsymbol{g}$, where $\boldsymbol{g}$ is the gradient of the loss. For the $\ell_\infty$-norm, the steepest descent direction is $\text{sign}(\boldsymbol{g})$ [15], and for $\ell_2$, it is $\boldsymbol{g}/\|\boldsymbol{g}\|_2$. For the $\ell_1$-norm, the steepest descent direction is the unit vector $\boldsymbol{e}$ with $e_{i*} = \text{sign}(g_{i*})$, for $i^* = \arg\max_i |g_i|$.

This yields an inefficient attack, as each iteration updates a single index of the perturbation $\boldsymbol{r}$. We thus design a new attack with finer control over the sparsity of an update step. For $q \in [0,1]$, let $P_q(|\boldsymbol{g}|)$ be the $q^{\text{th}}$ *percentile* of $|\boldsymbol{g}|$. We set $e_i = \text{sign}(g_i)$ if $|g_i| \geq P_q(|\boldsymbol{g}|)$ and $0$ otherwise, and normalize $\boldsymbol{e}$ to unit $\ell_1$-norm. For $q \gg 1/d$, we thus update many indices of $\boldsymbol{r}$ at once. We introduce another optimization to handle clipping, by ignoring gradient components where the update step cannot make progress (i.e., where $x_i + r_i \in \{0,1\}$ and $g_i$ points outside the domain). To project $\boldsymbol{r}$ onto an $\ell_1$-ball, we use an algorithm of Duchi et al. [10]. Algorithm 1 describes our attack. It outperforms the steepest descent attack as well as a recently proposed Frank-Wolfe algorithm for $\ell_1$-attacks [20] (see Appendix B). Our attack is competitive with the more expensive EAD attack [8] (see Appendix C).

## 4 Experiments

We use our new adversarial training schemes to measure the robustness trade-off on MNIST and CIFAR10.[1] MNIST is an interesting case-study as *distinct* models achieve strong robustness to different $\ell_p$ and spatial attacks[31, 11]. Despite the dataset's simplicity, we show that no single model achieves strong $\ell_\infty, \ell_1$ and $\ell_2$ robustness, and that new techniques are required to close this gap. The code used for all of our experiments can be found here: `https://github.com/ftramer/MultiRobustness`

**Training and evaluation setup.** We first use adversarial training to train models on a single perturbation type. For MNIST, we use $\ell_1(\epsilon = 10)$, $\ell_2(\epsilon = 2)$ and $\ell_\infty(\epsilon = 0.3)$. For CIFAR10 we use $\ell_\infty(\epsilon = \frac{4}{255})$ and $\ell_1(\epsilon = \frac{2000}{255})$. We also train on rotation-translation attacks with $\pm 3$px translations and $\pm 30°$ rotations as in [11]. We denote these models $\text{Adv}_1$, $\text{Adv}_2$, $\text{Adv}_\infty$, and $\text{Adv}_{\text{RT}}$. We then use the "max" and "avg" strategies from Section 3 to train models $\text{Adv}_{\text{max}}$ and $\text{Adv}_{\text{avg}}$ against multiple perturbations. We train once on all $\ell_p$-perturbations, and once on both $\ell_\infty$ and RT perturbations. We use the same CNN (for MNIST) and wide ResNet model (for CIFAR10) as Madry et al. [25]. Appendix A has more details on the training setup, and attack and training hyper-parameters.

We evaluate robustness of all models using multiple attacks: (1) we use *gradient-based attacks* for all $\ell_p$-norms, i.e., PGD [25] and our SLIDE attack with 100 steps and 40 restarts (20 restarts on CIFAR10), as well as Carlini and Wagner's $\ell_2$-attack [6] (C&W), and an $\ell_1$-variant—EAD [8];

Table 1: **Evaluation of MNIST models trained on $\ell_\infty, \ell_1$ and $\ell_2$ attacks (left) or $\ell_\infty$ and rotation-translation (RT) attacks (right).** Models $\text{Adv}_\infty$, $\text{Adv}_1$, $\text{Adv}_2$ and $\text{Adv}_{\text{RT}}$ are trained on a single attack, while $\text{Adv}_{\text{avg}}$ and $\text{Adv}_{\text{max}}$ are trained on multiple attacks using the "avg" and "max" strategies. The columns show a model's accuracy on individual perturbation types, on the union of them $(1 - \mathcal{R}^{\max}_{\text{adv}})$, and the average accuracy across them $(1 - \mathcal{R}^{\text{avg}}_{\text{adv}})$. The best results are in bold (at $95\%$ confidence). Results in red indicate gradient-masking, see Appendix C for a breakdown of all attacks.

| Model | Acc. | $\ell_\infty$ | $\ell_1$ | $\ell_2$ | $1 - \mathcal{R}^{\max}_{\text{adv}}$ | $1 - \mathcal{R}^{\text{avg}}_{\text{adv}}$ |
|---|---|---|---|---|---|---|
| Nat | **99.4** | 0.0 | 12.4 | 8.5 | 0.0 | 7.0 |
| $\text{Adv}_\infty$ | **99.1** | **91.1** | 12.1 | 11.3 | 6.8 | 38.2 |
| $\text{Adv}_1$ | 98.9 | 0.0 | **78.5** | 50.6 | 0.0 | 43.0 |
| $\text{Adv}_2$ | 98.5 | 0.4 | 68.0 | **71.8** | 0.4 | 46.7 |
| $\text{Adv}_{\text{avg}}$ | 97.3 | 76.7 | 53.9 | 58.3 | **49.9** | 63.0 |
| $\text{Adv}_{\text{max}}$ | 97.2 | 71.7 | 62.6 | 56.0 | **52.4** | **63.4** |

| Model | Acc. | $\ell_\infty$ | RT | $1 - \mathcal{R}^{\max}_{\text{adv}}$ | $1 - \mathcal{R}^{\text{avg}}_{\text{adv}}$ |
|---|---|---|---|---|---|
| Nat | **99.4** | 0.0 | 0.0 | 0.0 | 0.0 |
| $\text{Adv}_\infty$ | **99.1** | **91.4** | 0.2 | 0.2 | 45.8 |
| $\text{Adv}_{\text{RT}}$ | 99.3 | 0.0 | **94.6** | 0.0 | 47.3 |
| $\text{Adv}_{\text{avg}}$ | 99.2 | 88.2 | 86.4 | **82.9** | 87.3 |
| $\text{Adv}_{\text{max}}$ | 98.9 | 89.6 | 85.6 | **83.8** | **87.6** |

(2) to detect gradient-masking, we use *decision-based attacks*: the Boundary Attack [3] for $\ell_2$, the Pointwise Attack [31] for $\ell_1$, and the Boundary Attack++ [7] for $\ell_\infty$; (3) for spatial attacks, we use the optimal attack of [11] that enumerates all small rotations and translations. For unbounded attacks (C&W, EAD and decision-based attacks), we discard perturbations outside the $\ell_p$-ball.

For each model, we report accuracy on 1000 test points for: (1) individual perturbation types; (2) the union of these types, i.e., $1 - \mathcal{R}^{\text{avg}}_{\text{adv}}$; and (3) the average of all perturbation types, $1 - \mathcal{R}^{\text{avg}}_{\text{adv}}$. We briefly discuss the optimal error that can be achieved if there is no robustness trade-off. For perturbation sets $S_1, \ldots S_n$, let $\mathcal{R}_1, \ldots, \mathcal{R}_n$ be the optimal risks achieved by distinct models. Then, a single model can at best achieve risk $\mathcal{R}_i$ for each $S_i$, i.e., $\text{OPT}(\mathcal{R}^{\text{avg}}_{\text{adv}}) = \frac{1}{n}\sum_{i=1}^{n}\mathcal{R}_i$. If the errors are fully correlated, so that a maximal number of inputs admit *no* attack, we have $\text{OPT}(\mathcal{R}^{\max}_{\text{adv}}) = \max\{\mathcal{R}_1, \ldots, \mathcal{R}_n\}$. Our experiments show that these optimal error rates are not achieved.

**Results on MNIST.** Results are in Table 1. The left table is for the union of $\ell_p$-attacks, and the right table is for the union of $\ell_\infty$ and RT attacks. In both cases, the multi-perturbation training strategies "succeed", in that models $\text{Adv}_{\text{avg}}$ and $\text{Adv}_{\text{max}}$ achieve higher multi-perturbation accuracy than any of the models trained against a single perturbation type.

The results for $\ell_\infty$ and RT attacks are promising, although the best model $\text{Adv}_{\text{max}}$ only achieves $1 - \mathcal{R}^{\max}_{\text{adv}} = 83.8\%$ and $1 - \mathcal{R}^{\text{avg}}_{\text{adv}} = 87.6\%$, which is far less than the optimal values, $1 - \text{OPT}(\mathcal{R}^{\max}_{\text{adv}}) = \min\{91.4\%, 94.6\%\} = 91.4\%$ and $1 - \text{OPT}(\mathcal{R}^{\text{avg}}_{\text{adv}}) = (91.4\% + 94.6\%)/2 = 93\%$. Thus, these models do exhibit some form of the robustness trade-off analyzed in Section 2.

The $\ell_p$ results are surprisingly mediocre and re-raise questions about whether MNIST can be considered "solved" from a robustness perspective. Indeed, while training *separate* models to resist $\ell_1, \ell_2$ or $\ell_\infty$ attacks works well, resisting all attacks simultaneously fails. This agrees with the results of Schott et al. [31], whose models achieve either high $\ell_\infty$ or $\ell_2$ robustness, but not both simultaneously. We show that in our case, this lack of robustness is partly due to gradient masking.

**First-order adversarial training and gradient masking on MNIST.** The model $\text{Adv}_\infty$ is not robust to $\ell_1$ and $\ell_2$-attacks. This is unsurprising as the model was only trained on $\ell_\infty$-attacks. Yet, comparing the model's accuracy against multiple types of $\ell_1$ and $\ell_2$ attacks (see Appendix C) reveals a more curious phenomenon: $\text{Adv}_\infty$ has high accuracy against *first-order* $\ell_1$ and $\ell_2$-attacks such as PGD, but is broken by decision-free attacks. This is an indication of gradient-masking [27, 40, 1].

This issue had been observed before [31, 23], but an explanation remained illusive, especially since $\ell_\infty$-PGD does not appear to suffer from gradient masking (see [25]). We explain this phenomenon by inspecting the learned features of model $\text{Adv}_\infty$, as in [25]. We find that the model's first layer learns threshold filters $\boldsymbol{z} = \text{ReLU}(\alpha \cdot (\boldsymbol{x} - \epsilon))$ for $\alpha > 0$. As most pixels in MNIST are zero, most of the $z_i$ cannot be activated by an $\epsilon$-bounded $\ell_\infty$-attack. The $\ell_\infty$-PGD thus optimizes a smooth (albeit flat) loss function. In contrast, $\ell_1$- and $\ell_2$-attacks can move a pixel $x_i = 0$ to $\hat{x}_i > \epsilon$ thus activating $z_i$, but have no gradients to rely on (i.e, $dz_i/dx_i = 0$ for any $x_i \leq \epsilon$). Figure 3 in Appendix D shows that the model's loss resembles a step-function, for which first-order attacks such as PGD are inadequate.

Note that training against first-order $\ell_1$ or $\ell_2$-attacks directly (i.e., models $\text{Adv}_1$ and $\text{Adv}_2$ in Table 1), seems to yield genuine robustness to these perturbations. This is surprising in that, because of gradient

Table 2: **Evaluation of CIFAR10 models trained against $\ell_\infty$ and $\ell_1$ attacks (left) or $\ell_\infty$ and rotation-translation (RT) attacks (right).** Models $\text{Adv}_\infty$, $\text{Adv}_1$ and $\text{Adv}_{RT}$ are trained against a single attack, while $\text{Adv}_{avg}$ and $\text{Adv}_{max}$ are trained against two attacks using the "avg" and "max" strategies. The columns show a model's accuracy on individual perturbation types, on the union of them $(1 - \mathcal{R}_{adv}^{max})$, and the average accuracy across them $(1 - \mathcal{R}_{adv}^{avg})$. The best results are in bold (at $95\%$ confidence). A breakdown of all $\ell_1$ attacks is in Appendix C.

| Model | Acc. | $\ell_\infty$ | $\ell_1$ | $1 - \mathcal{R}_{adv}^{max}$ | $1 - \mathcal{R}_{adv}^{avg}$ |
|---|---|---|---|---|---|
| Nat | **95.7** | 0.0 | 0.0 | 0.0 | 0.0 |
| $\text{Adv}_\infty$ | 92.0 | **71.0** | 16.4 | 16.4 | 44.9 |
| $\text{Adv}_1$ | 90.8 | 53.4 | **66.2** | 53.1 | 60.0 |
| $\text{Adv}_{avg}$ | 91.1 | 64.1 | 60.8 | **59.4** | **62.5** |
| $\text{Adv}_{max}$ | 91.2 | 65.7 | 62.5 | **61.1** | **64.1** |

| Model | Acc. | $\ell_\infty$ | RT | $1 - \mathcal{R}_{adv}^{max}$ | $1 - \mathcal{R}_{adv}^{avg}$ |
|---|---|---|---|---|---|
| Nat | **95.7** | 0.0 | 5.9 | 0.0 | 3.0 |
| $\text{Adv}_\infty$ | 92.0 | **71.0** | 8.9 | 8.7 | 40.0 |
| $\text{Adv}_{RT}$ | 94.9 | 0.0 | **82.5** | 0.0 | 41.3 |
| $\text{Adv}_{avg}$ | 93.6 | 67.8 | 78.2 | **65.2** | **73.0** |
| $\text{Adv}_{max}$ | 93.1 | 69.6 | 75.2 | **65.7** | **72.4** |

Table 3: **Evaluation of affine attacks.** For models trained with the "max" strategy, we evaluate against attacks from a union $S_U$ of perturbation sets, and against an affine adversary that interpolates between perturbations. Examples of affine attacks are in Figure 4.

| Dataset | Attacks | acc. on $S_U$ | acc. on $S_{affine}$ |
|---|---|---|---|
| MNIST | $\ell_\infty$ & RT | 83.8 | 62.6 |
| CIFAR10 | $\ell_\infty$ & RT | 65.7 | 56.0 |
| CIFAR10 | $\ell_\infty$ & $\ell_1$ | 61.1 | 58.0 |

masking, model $\text{Adv}_\infty$ actually achieves lower training loss against first-order $\ell_1$ and $\ell_2$-attacks than models $\text{Adv}_1$ and $\text{Adv}_2$. That is, $\text{Adv}_1$ and $\text{Adv}_2$ converged to sub-optimal local minima of their respective training objectives, yet these minima generalize much better to stronger attacks.

The models $\text{Adv}_{avg}$ and $\text{Adv}_{max}$ that are trained against $\ell_\infty, \ell_1$ and $\ell_2$-attacks also learn to use thresholding to resist $\ell_\infty$-attacks while spuriously masking gradient for $\ell_1$ and $\ell_2$-attacks. This is evidence that, unlike previously thought [41], training against a strong first-order attack (such as PGD) can cause the model to minimize its training loss via gradient masking. To circumvent this issue, alternatives to first-order adversarial training seem necessary. Potential (costly) approaches include training on gradient-free attacks, or extending certified defenses [28, 42] to multiple perturbations. Certified defenses provide provable bounds that are much weaker than the robustness attained by adversarial training, and certifying multiple perturbation types is likely to exacerbate this gap.

**Results on CIFAR10.** The left table in Table 2 considers the union of $\ell_\infty$ and $\ell_1$ perturbations, while the right table considers the union of $\ell_\infty$ and RT perturbations. As on MNIST, the models $\text{Adv}_{avg}$ and $\text{Adv}_{max}$ achieve better multi-perturbation robustness than any of the models trained on a single perturbation, but fail to match the optimal error rates we could hope for. For $\ell_1$ and $\ell_\infty$-attacks, we achieve $1 - \mathcal{R}_{adv}^{max} = 61.1\%$ and $1 - \mathcal{R}_{adv}^{avg} = 64.1\%$, again significantly below the optimal values, $1 - \text{OPT}(\mathcal{R}_{adv}^{max}) = \min\{71.0\%, 66.2\%\} = 66.2\%$ and $1 - \text{OPT}(\mathcal{R}_{adv}^{avg}) = (71.0\% + 66.2\%)/2 = 68.6\%$. The results for $\ell_\infty$ and RT attacks are qualitatively and quantitatively similar. [2]

Interestingly, models $\text{Adv}_{avg}$ and $\text{Adv}_{max}$ achieve $100\%$ *training accuracy*. Thus, multi-perturbation robustness increases the *adversarial generalization gap* [30]. These models might be resorting to more memorization because they fail to find features robust to both attacks.

**Affine Adversaries.** Finally, we evaluate the affine attacks introduced in Section 2.5. These attacks take affine combinations of two perturbation types, and we apply them on the models $\text{Adv}_{max}$ (we omit the $\ell_p$-case on MNIST due to gradient masking). To compound $\ell_\infty$ and $\ell_1$-noise, we devise an attack that updates both perturbations in alternation. To compound $\ell_\infty$ and RT attacks, we pick random rotation-translations (with $\pm 3\beta$px translations and $\pm 30\beta°$ rotations), apply an $\ell_\infty$-attack with budget $(1 - \beta)\epsilon$ to each, and retain the worst example.

The results in Table 3 match the predictions of our formal analysis: (1) affine combinations of $\ell_p$ perturbations are no stronger than their union. This is expected given Claim 3 and prior observations that neural networks are close to linear near the data [15, 29]; (2) combining of $\ell_\infty$ and RT attacks does yield a stronger attack, as shown in Theorem 4. This demonstrates that robustness to a union of perturbations can still be insufficient to protect against more complex combinations of perturbations.

## 5 Discussion and Open Problems

Despite recent success in defending ML models against some perturbation types [25, 11, 31], extending these defenses to multiple perturbations unveils a clear robustness trade-off. This tension may be rooted in its unconditional occurrence in natural and simple distributions, as we proved in Section 2.

Our new adversarial training strategies fail to achieve competitive robustness to more than one attack type, but narrow the gap towards multi-perturbation robustness. We note that the optimal risks $\mathcal{R}_{\text{adv}}^{\text{max}}$ and $\mathcal{R}_{\text{adv}}^{\text{avg}}$ that we achieve are very close. Thus, for most data points, the models are either robust to all perturbation types or none of them. This hints that some points (sometimes referred to as *prototypical examples* [4, 36]) are inherently easier to classify robustly, regardless of the perturbation type.

We showed that first-order adversarial training for multiple $\ell_p$-attacks suffers from gradient masking on MNIST. Achieving better robustness on this simple dataset is an open problem. Another challenge is reducing the cost of our adversarial training strategies, which scale linearly in the number of perturbation types. Breaking this linear dependency requires efficient techniques for finding perturbations in a union of sets, which might be hard for sets with near-empty intersection (e.g., $\ell_\infty$ and $\ell_1$-balls). The cost of adversarial training has also be reduced by merging the inner loop of a PGD attack and gradient updates of the model parameters [34, 44], but it is unclear how to extend this approach to a union of perturbations (some of which are not optimized using PGD, e.g., rotation-translations).

Hendrycks and Dietterich [17], and Geirhos et al. [13] recently measured robustness of classifiers to multiple common (i.e., non-adversarial) image corruptions (e.g., random image blurring). In that setting, they also find that different classifiers achieve better robustness to some corruptions, and that no single classifier achieves the highest accuracy under all forms. The interplay between multi-perturbation robustness in the adversarial and common corruption case is worth further exploration.

## Footnotes

[1]Kang et al. [20] recently studied the transfer between $\ell_\infty, \ell_1$ and $\ell_2$-attacks for adversarially trained models on ImageNet. They show that models trained on one type of perturbation are not robust to others, but they do not attempt to train models against multiple attacks simultaneously.

[2] An interesting open question is why the model $\text{Adv}_{avg}$ trained on $\ell_\infty$ and RT attacks does not attain optimal average robustness $\mathcal{R}_{adv}^{avg}$. Indeed, on CIFAR10, detecting the RT attack of [11] is easy, due to the black in-painted pixels in a transformed image. The following "ensemble" model thus achieves optimal $\mathcal{R}_{adv}^{avg}$ (but not necessarily optimal $\mathcal{R}_{adv}^{max}$): on input $\hat{x}$, return $\text{Adv}_{RT}(\hat{x})$ if there are black in-painted pixels, otherwise return $\text{Adv}_\infty(\hat{x})$. The fact that model $\text{Adv}_{avg}$ did not learn such a function might hint at some limitation of adversarial training.

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
