[Supplementary Material]

## A Experimental Setup

**MNIST.** We use the CNN model from Madry et al. [24] and train for 10 epochs with Adam and a learning rate of $10^{-3}$ reduced to $10^{-4}$ after 5 epochs (batch size of 100). To accelerate convergence, we train against a weaker adversary in the first epoch (with $1/3$ of the perturbation budget). For training, we use PGD with 40 iterations for $\ell_\infty$ and 100 iterations for $\ell_1$ and $\ell_2$. For rotation-translations, we use the attack from [10] that picks the worst of 10 random rotation-translations.

**CIFAR10.** We use the same wide ResNet model as [24]. We train for 80k steps of gradient descent with batch size 128 (205 epochs). When using the "avg" strategy for wide ResNet models, we had to halve the batch size to avoid overflowing the GPU's memory. We accordingly doubled the number of training steps and learning rate schedule. We use a learning rate of 0.1 decayed by a factor 10 after 40k and 60k steps, a momentum of 0.9, and weight decay of 0.0002. Except for the RT attack, we use standard data augmentation with random padding, cropping and horizontal flipping (see [10] for details). We extract 1,000 points from the CIFAR10 test as a validation set for early-stopping.

For training, we use PGD with 10 iterations for $\ell_\infty$, and 20 iterations for $\ell_1$. [4] For rotation-translations, we also use the attack from [10] that trains on the worst of 10 randomly chosen rotation-translations.

## B Performance of the Sparse $\ell_1$-Descent Attack

In Figure 2, we compare the performance of our new Sparse $\ell_1$-Descent Attack (SLIDE) for different choices of gradient sparsity. We also compare to the standard PGD attack with the steepest-descent update rule, as well as a recent attack proposed in [19] that adapts the Frank-Wolfe optimization algorithm for finding $\ell_1$-bounded adversarial examples. As we explained in Section 3, we expect our attack to outperform PGD as the steepest-descent vector is too sparse in the $\ell_1$-case, and we indeed observe a significant improvement by choosing denser updates.

The subpar performance of the Frank-Wolfe algorithm is also intriguing. We believe it is due to the attack's linearly decreasing step-size (the $k^{\text{th}}$ iteration has a step-size of $O(1/k)$, see [19] for details). While this choice is appropriate for optimizing convex functions, in the non-convex case it overly emphasizes the first steps of the attack, which intuitively should increase the likelihood of landing in a local minima.

Figure 2: **Performance of the Sparse $\ell_1$-Descent Attack on MNIST (left) and CIFAR10 (right) for different choices of descent directions.** We run the attack for up to 1,000 steps and plot the evolution of the cross-entropy loss, for an undefended model. We vary the sparsity of the gradient updates (controlled by the parameter $q$), and compare to the standard PGD attack that uses the steepest descent vector, as well as the Frank-Wolfe $\ell_1$-attack from [19]. For appropriate $q$, our attack vastly outperforms PGD and Frank-Wolfe.

# C   Breakdown of $\ell_p$-Attacks on Adversarially Trained Models

Tables 4 and 5 below give a more detailed breakdown of each model's accuracy against each $\ell_p$ attack we considered. For each model and attack, we evaluate the attack on 1,000 test points and report the accuracy. For each individual perturbation type (i.e., $\ell_\infty, \ell_1, \ell_2$), we further report the accuracy obtained by choosing the worst attack for each input. Finally, we report the accuracy against the union of all attacks $(1 - \mathcal{R}_{adv}^{max})$ as well as the average accuracy across perturbation types $(1 - \mathcal{R}_{adv}^{avg})$.

Table 4: **Breakdown of all attacks on MNIST models.** For $\ell_\infty$, we use PGD and the Boundary Attack++ (BAPP) [7]. For $\ell_1$, we use our new Sparse $\ell_1$-Descent Attack (SLIDE), EAD [8] and the Pointwise Attack (PA) [30]. For $\ell_2$, we use PGD, C&W [6] and the Boundary Attack (BA) [3].

| Model | Acc. | $\ell_\infty$ | | | $\ell_1$ | | | | $\ell_2$ | | | | $1 - \mathcal{R}_{adv}^{max}$ | $1 - \mathcal{R}_{adv}^{avg}$ |
|---|---|---|---|---|---|---|---|---|---|---|---|---|---|---|
| | | PGD | BAPP | All $\ell_\infty$ | SLIDE | EAD | PA | All $\ell_1$ | PGD | C&W | BA | All $\ell_2$ | | |
| Nat | **99.4** | 0.0 | 13.0 | 0.0 | 13.0 | 18.8 | 72.1 | 12.4 | 11.0 | 10.4 | 31.0 | 8.5 | 0.0 | 7.0 |
| $\text{Adv}_\infty$ | **99.1** | 91.1 | 98.5 | **91.1** | 66.9 | 58.4 | 15.0 | 12.1 | 78.1 | 78.4 | 14.0 | 11.3 | 6.8 | 38.2 |
| $\text{Adv}_1$ | 98.9 | 0.0 | 43.5 | 0.0 | 78.6 | 81.0 | 91.6 | **78.5** | 53.0 | 52.0 | 69.7 | 50.6 | 0.0 | 43.0 |
| $\text{Adv}_2$ | 98.5 | 0.4 | 78.5 | 0.4 | 70.4 | 69.3 | 89.7 | 68.0 | 74.7 | 74.5 | 81.7 | **71.8** | 0.4 | 46.7 |
| $\text{Adv}_{avg}$ | 97.3 | 76.7 | 98.0 | 76.7 | 66.3 | 62.4 | 68.6 | 53.9 | 77.7 | 72.3 | 64.6 | 58.3 | **49.9** | **63.0** |
| $\text{Adv}_{max}$ | 97.2 | 71.7 | 98.5 | 71.7 | 72.1 | 70.0 | 69.6 | 62.6 | 75.7 | 71.8 | 59.7 | 56.0 | **52.4** | **63.4** |

Table 5: **Breakdown of all attacks on CIFAR10 models.** For $\ell_\infty$, we use PGD. For $\ell_1$, we use our new Sparse $\ell_1$-descent attack (SLIDE), EAD [8] and the Pointwise Attack (PA) [30].

| Model | Acc. | $\ell_\infty$ | | $\ell_1$ | | | | $1 - \mathcal{R}_{adv}^{max}$ | $1 - \mathcal{R}_{adv}^{avg}$ |
|---|---|---|---|---|---|---|---|---|---|
| | | PGD | All $\ell_\infty$ | SLIDE | EAD | PA | All $\ell_1$ | | |
| Nat | **95.7** | 0.0 | 0.0 | 0.2 | 0.0 | 29.6 | 0.0 | 0.0 | 0.0 |
| $\text{Adv}_\infty$ | 92.0 | 71.0 | **71.0** | 19.4 | 17.6 | 52.7 | 16.4 | 16.4 | 44.9 |
| $\text{Adv}_1$ | 90.8 | 53.4 | 53.4 | 66.6 | 66.6 | 84.7 | **66.2** | 53.1 | 60.0 |
| $\text{Adv}_{avg}$ | 91.1 | 64.1 | 64.1 | 61.1 | 61.5 | 81.7 | 60.8 | **59.4** | **62.5** |
| $\text{Adv}_{max}$ | 91.2 | 65.7 | 65.7 | 63.1 | 63.0 | 83.4 | 62.5 | **61.1** | **64.1** |

# D   Gradient Masking as a Consequence of $\ell_\infty$-Robustness on MNIST.

Multiple works have reported on a curious phenomenon that affects the $\ell_\infty$-adversarially trained model of Madry et al. [24] on MNIST. This model achieves strong robustness to the $\ell_\infty$ attacks it was trained on, as one would expect. Yet, on other $\ell_p$-norms (e.g., $\ell_1$ [8, 30] and $\ell_2$ [22, 30]), its robustness is no better—or even worse—than for an undefended model. Some authors have referred to this effect as *overfitting*, a somewhat unfair assessment of the work of [24], as their model actually achieves exactly what it was trained to do—namely resist $\ell_\infty$-bounded attacks. Moreover, as our theoretical results suggest, this trade-off may be inevitable (a similar point was made in [20]).

The more intriguing aspect of Madry et al.'s MNIST model is that, when attacked by $\ell_1$ or $\ell_2$ adversaries, first-order attacks are sub-optimal. This was previously observed in [30] and in [22], where decision-based or second-order attacks vastly outperformed gradient descent for finding $\ell_1$ or $\ell_2$ adversarial examples. Li et al. [22] argue that this effect is due to the gradients of the adversarially trained model having much smaller magnitude than in a standard model. Yet, this fails to explain why first-order attacks appear to be optimal in the $\ell_\infty$-norm that the model was trained against.

A natural explanation for this discrepancy follows from an inspection of the robust model's first layer (as done in [24]). All kernels of the model's first convolutional layer have very small norm, except for three kernels that have a single large weight. This reduces the convolution to a thresholding filter, which we find to be of one of two forms: either $\text{ReLU}(\alpha \cdot (x - 0.3))$ or $\text{ReLU}(\alpha \cdot (x - 0.7))$ for constant $\alpha > 0$.[5] Thus, the model's first layer forms a piece-wise function with three distinct regimes, depending on the value of an input pixel $x_i$: (1) for $x_i \in [0, 0.3]$, the output is only influenced by the

Figure 3: **Gradient masking in an $\ell_\infty$-adversarially trained model on MNIST, evaluated against $\ell_1$-attacks (left) and $\ell_2$-attacks (right).** The model is trained against an $\ell_\infty$-PGD adversary with $\epsilon = 0.3$. For a randomly chosen data point $x$, we compute an adversarial perturbation $r_{\mathrm{PGD}}$ using PGD and $r_{\mathrm{GF}}$ using a gradient-free attack. The left plot is for $\ell_1$-attacks with $\epsilon = 10$ and the right plot is for $\ell_2$-attacks with $\epsilon = 2$. The plots display the loss on points of the form $\hat{x} := x + \alpha \cdot r_{\mathrm{PGD}} + \beta \cdot r_{\mathrm{GF}}$, for $\alpha, \beta \in [0, \epsilon]$. The loss surface behaves like a step-function, and gradient-free attacks succeed in finding adversarial examples where first-order methods failed.

463 low-weight kernels. For $x_i \in [0.3, 1]$, the $\mathrm{ReLU}(\alpha \cdot (x - 0.3))$ filters become active, and override the
464 signal from the low-weight kernels. For $x_i \in [0.7, 1]$, the $\mathrm{ReLU}(\alpha \cdot (x - 0.7))$ filters are also active.

465 As most MNIST pixels are in $\{0, 1\}$, $\ell_\infty$-attacks operate in a regime where most perturbed pixels
466 are in $[0, 0.3] \cup [0.7, 1]$. The model's large-weight ReLUs thus never transition between active and
467 inactive, which leads to a smooth, albeit flat loss that first-order methods navigate effectively.

468 For $\ell_1$ and $\ell_2$ attacks however, one would expect some of the ReLUs to be flipped as the attacks can
469 make changes larger that $0.3$ to some pixels. Yet, as most MNIST pixels are $0$ (the digit's background),
470 nearly all large-weight ReLUs start out inactive, with gradients equal to zero. A first-order adversary
471 thus has no information on which pixels to focus the perturbation budget on.

472 Decision-based attacks sidestep this issue by disregarding gradients entirely. Figure 3 shows two
473 examples of input points where a decision-based attack (Pointwise Attack for $\ell_1$ [30] and Boundary
474 Attack for $\ell_2$ [3]) finds an adversarial example in a direction that is orthogonal to the one explored by
475 PGD. The loss surface exhibits sharp thresholding steps, as predicted by our analysis.

476 When we explicitly train against first-order $\ell_1$ or $\ell_2$ adversaries (models $\mathrm{Adv}_1$ and $\mathrm{Adv}_2$ in Table 1,
477 left), the resulting model is robust (at least empirically) to $\ell_1$ or $\ell_2$ attacks. Note that model $\mathrm{Adv}_\infty$
478 actually achieves higher robustness to $\ell_2$-PGD attacks than $\mathrm{Adv}_2$ (due to gradient-masking). Thus,
479 the $\mathrm{Adv}_2$ model converged to a *sub-optimal* local minima of its first-order adversarial training
480 procedure (i.e., learning the same thresholding mechanism as $\mathrm{Adv}_\infty$ would yield lower loss). Yet,
481 this sub-optimal local minima generalizes much better to other $\ell_2$ attacks.

482 Models trained against $\ell_\infty, \ell_1$ and $\ell_2$ attacks (i.e., $\mathrm{Adv}_{\mathrm{all}}$ and $\mathrm{Adv}_{\mathrm{max}}$) in Table 1, left) also learn to
483 use thresholding to achieve robustness to $\ell_\infty$ attacks, while masking gradients for $\ell_1$ and $\ell_2$ attacks.

## E Examples of Affine Combinations of Perturbations

In Figure 4, we display examples of $\ell_1$, $\ell_\infty$ and rotation-translation attacks on MNIST and CIFAR10, as well as affine attacks that interpolate between two attack types.

Figure 4: **Adversarial examples for $\ell_\infty$, $\ell_1$ and rotation-translation (RT) attacks, and affine combinations thereof.** The first column in each subplot shows clean images. The following five images in each row linearly interpolate between two attack types, as described in Section 2.5. Images marked in red are mis-classified by a model trained against both types of perturbations. Note that there are examples for which combining a rotation-translation and $\ell_\infty$-attack is stronger than either perturbation type individually.

## F Proof of Theorem 1 (Robustness trade-off between $\ell_\infty$ and $\ell_1$- norms)

Our proof follows a similar structure to the proof of Theorem 2.1 in [39], although the analysis is slightly simplified in our case as we are comparing two perturbation models, an $\ell_\infty$-bounded one and an $\ell_1$-bounded one, that are essentially orthogonal to each other. With a perturbation of size $\epsilon = 2\eta$, the $\ell_\infty$-bounded noise can "flip" the distribution of the features $x_1, \ldots, x_d$ to reflect the opposite label, and thus destroy any information that a classifier might extract from those features. On the other side, an $\ell_1$-bounded perturbation with $\epsilon = 2$ can flip the distribution of $x_0$. By sacrificing some features, a classifier can thus achieve some robustness to either $\ell_\infty$ *or* $\ell_1$ noise, but never to both simultaneously.

For $y \in \{-1, +1\}$, let $\mathcal{G}^y$ be the distribution over feature $x_0$ conditioned on the value of $y$. Similarly, let $\mathcal{H}^y$ be the conditional distribution over features $x_1, \ldots, x_d$. Consider the following perturbations: $\boldsymbol{r}_\infty = [0, -2y\eta, \ldots, -2y\eta]$ has small $\ell_\infty$-norm, and $\boldsymbol{r}_1 = [-2x_0, 0, \ldots, 0]$ has small $\ell_1$-norm. The $\ell_\infty$ perturbation can change $\mathcal{H}^y$ to $\mathcal{H}^{-y}$, while the $\ell_1$ perturbation can change $\mathcal{G}^y$ to $\mathcal{G}^{-y}$.

Let $f(\boldsymbol{x})$ be any classifier from $\mathbb{R}^{d+1}$ to $\{-1, +1\}$ and define:

$$p_{+-} = \Pr_{\boldsymbol{x} \sim (\mathcal{G}^{+1}, \mathcal{H}^{-1})}[f(\boldsymbol{x}) = +1], \qquad p_{-+} = \Pr_{\boldsymbol{x} \sim (\mathcal{G}^{-1}, \mathcal{H}^{+1})}[f(\boldsymbol{x}) = +1].$$

The accuracy of $f$ against the $\boldsymbol{r}_\infty$ perturbation is given by:

$$\Pr[f(\boldsymbol{x} + \boldsymbol{r}_\infty) = y] = \Pr[y = +1] \cdot p_{+-} + \Pr[y = -1] \cdot (1 - p_{-+}) = \frac{1}{2} \cdot (1 + p_{+-} - p_{-+}).$$

Similarly, the accuracy of $f$ against the $\boldsymbol{r}_1$ perturbation is:

$$\Pr[f(\boldsymbol{x} + \boldsymbol{r}_1) = y] = \Pr[y = +1] \cdot p_{-+} + \Pr[y = -1] \cdot (1 - p_{+-}) = \frac{1}{2} \cdot (1 + p_{-+} - p_{+-}).$$

Combining these, we get $\Pr[f(\boldsymbol{x} + \boldsymbol{r}_\infty) = y] + \Pr[f(\boldsymbol{x} + \boldsymbol{r}_1) = y] = 1$.

As $\boldsymbol{r}_\infty$ and $\boldsymbol{r}_1$ are two specific $\ell_\infty$- and $\ell_1$-bounded perturbations, the above is an upper-bound on the accuracy that $f$ achieves against worst-case perturbation within the prescribed noise models, which concludes the proof.

$\square$

## G   Proof of Theorem 2 (Robustness trade-off between $\ell_\infty$ and spatial perturbations)

The proof of this theorem follows a similar blueprint to the proof of Theorem 1. Recall that an $\ell_\infty$ perturbation with $\epsilon = 2\eta$ can flip the distribution of the features $x_1, \ldots, x_n$ to reflect an opposite label $y$. The tricky part of the proof is to show that a small rotation or translation can flip the distribution of $x_0$ to the opposite label, without affecting the marginal distribution of the other features too much.

Recall that we model rotations and translations as picking a permutation $\pi$ from some fixed set $\Pi$ of permutations over the indices in $\boldsymbol{x}$, with the constraint that feature $x_0$ be moved to at most $N$ different positions for all $\pi \in \Pi$.

We again define $\mathcal{G}^y$ as the distribution of $x_0$ conditioned on $y$, and $\mathcal{H}^y$ for the distribution of $x_1, \ldots, x_d$. We know that a small $\ell_\infty$-perturbation can transform $\mathcal{H}^y$ into $\mathcal{H}^{-y}$. Our goal is to show that a rotation-translation adversary can change $(\mathcal{G}^y, \mathcal{H}^y)$ into a distribution that is very close to $(\mathcal{G}^{-y}, \mathcal{H}^y)$. The result of the theorem then follows by arguing that no binary classifier $f$ can distinguish, with high accuracy, between $\ell_\infty$-perturbed examples with label $y$ and rotated examples with label $-y$ (and vice versa).

We first describe our proof idea at a high level. We define an intermediate "hybrid" distribution $\mathcal{Z}^y$ where all $d+1$ features are i.i.d $N(y\eta, 1)$ (that is, $x_0$ now has the same distribution as the other weakly-correlated features). The main step in the proof is to show that for samples from either $(\mathcal{G}^y, \mathcal{H}^y)$ or $(\mathcal{G}^{-y}, \mathcal{H}^y)$, a random rotation-translation yields a distribution that is very close (in total variation) to $\mathcal{Z}^y$. From this, we then show that there exists an adversary that applies two rotations or translations in a row, to first transform samples from $(\mathcal{G}^y, \mathcal{H}^y)$ into samples close to $\mathcal{Z}^y$, and then transform those samples into ones that are close to $(\mathcal{G}^{-y}, \mathcal{H}^y)$.

We will need a standard version of the Berry-Esseen theorem, stated hereafter for completeness.

**Theorem 5** (Berry-Esseen [2]). *Let $X_1, \ldots, X_n$ be independent random variables with $\mathbb{E}[X_i] = \mu_i$, $\mathbb{E}[X_i^2] = \sigma_i^2 > 0$, and $\mathbb{E}[|X_i|^3] = \rho_i < \infty$, where the $\mu_i, \sigma_i$ and $\rho_i$ are constants independent of $n$. Let $S_n = X_1 + \cdots + X_n$, with $F_n(x)$ the CDF of $S_n$ and $\Phi(x)$ the CDF of the standard normal distribution. Then,*

$$\sup_{x \in \mathbb{R}} \left| F_n(x) - \Phi\left( \frac{x - \mathbb{E}[S_n]}{\sqrt{\operatorname{Var}[S_n]}} \right) \right| = O(1/\sqrt{n}) \,.$$

For distributions $\mathcal{P}, \mathcal{Q}$, let $\Delta_{\mathrm{TV}}(\mathcal{P}, \mathcal{Q})$ denote their total-variation distance. The below lemma is the main technical result we need, and bounds the total variation between a multivariate Gaussian $\mathcal{P}$ and a special mixture of multivariate Gaussians $\mathcal{Q}$.

**Lemma 6.** *For $k > 1$, let $\mathcal{P}$ be a $k$-dimensional Gaussians with mean $\boldsymbol{\mu}_P = [\lambda_P, \ldots, \lambda_P]$ and identity covariance. For all $i \in [k]$, let $\mathcal{Q}_i$ be a multivariate Gaussian with mean $\boldsymbol{\mu}_i$ and diagonal covariance $\boldsymbol{\Sigma}_i$ where $(\boldsymbol{\mu}_i)_j = \begin{cases} \lambda_Q & \text{if } i = j \\ \lambda_P & \text{otherwise} \end{cases}$ and $(\boldsymbol{\Sigma}_i)_{(j,j)} = \begin{cases} \sigma_Q^2 & \text{if } i = j \\ 1 & \text{otherwise} \end{cases}$.*

*Define $\mathcal{Q}$ as a mixture distribution of the $\mathcal{Q}_1, \ldots, \mathcal{Q}_k$ with probabilities $1/k$. Assuming that $\lambda_P, \lambda_Q, \sigma_Q$ are constants independent of $k$, we have $\Delta_{TV}(\mathcal{P}, \mathcal{Q}) = O(1/\sqrt{k})$.*

*Proof.* [6] Let $p(\boldsymbol{x})$ and $q(\boldsymbol{x})$ denote, respectively, the pdfs of $\mathcal{P}$ and $\mathcal{Q}$. Note that $q(\boldsymbol{x}) = \sum_{i=1}^k \frac{1}{k} q_i(\boldsymbol{x})$, where $q_i(\boldsymbol{x})$ is the pdf of $\mathcal{Q}_i$. We first compute:

$$q(\boldsymbol{x}) = \sum_{i=1}^k \frac{1}{k} \frac{1}{\sqrt{(2\pi)^k \cdot |\boldsymbol{\Sigma}_i|}} \cdot e^{-\frac{1}{2}(\boldsymbol{x} - \boldsymbol{\mu}_i)^T \boldsymbol{\Sigma}_i^{-1}(\boldsymbol{x} - \boldsymbol{\mu}_i)}$$

$$= \frac{e^{-\frac{1}{2}(\boldsymbol{x} - \boldsymbol{\mu}_P)^T(\boldsymbol{x} - \boldsymbol{\mu}_P)}}{\sqrt{(2\pi)^k}} \cdot \frac{1}{k \cdot \sigma_Q^2} \cdot \sum_{i=1}^k e^{-\frac{1}{2}t(x_i)}$$

$$= p(\boldsymbol{x}) \cdot \frac{1}{k \cdot \sigma_Q^2} \cdot \sum_{i=1}^k e^{-\frac{1}{2}t(x_i)} \,,$$

where

$$t(x_i) := (\sigma_Q^{-2} - 1)x_i^2 - (2\lambda_Q \sigma_Q^{-2} - 2\lambda_P)x_i + (\lambda_Q^2 \sigma_Q^{-2} - \lambda_P^2) \,. \tag{3}$$

Thus we have that

$$q(\boldsymbol{x}) < p(\boldsymbol{x}) \quad \Longleftrightarrow \quad \frac{1}{k \cdot \sigma_Q^2} \cdot \sum_{i=1}^{k} e^{-\frac{1}{2}t(x_i)} < 1 \,.$$

The total-variation distance between $\mathcal{P}$ and $\mathcal{Q}$ is then $\Delta_{\mathrm{TV}}(\mathcal{P}, \mathcal{Q}) = p_1 - p_2$, where

$$p_1 := \Pr\left[S_k < k \cdot \sigma_Q^2\right] \,, \quad p_2 := \Pr\left[T_k < k \cdot \sigma_Q^2\right] \,, \tag{4}$$

$$S_k := \sum_{i=1}^{k} U_i \,, \quad T_k := S_{k-1} + V_k \,, \quad U_i := e^{-\frac{1}{2}t(Z_i)} \,, \quad V_n := e^{-\frac{1}{2}t(W_n)} \,,$$

and the $Z_i \sim \mathcal{N}(\lambda_P, 1)$, $W_n \sim \mathcal{N}(\lambda_Q, \sigma_Q^2)$ and all the $Z_i$ and $W_n$ are mutually independent.

It is easy to verify that $\mathbb{E}[U_i] = \sigma_Q^2$, $\mathrm{Var}[U_i] = O(1)$, $\mathbb{E}[U_i^3] = O(1)$, $\mathbb{E}[W_n] = O(1)$, $\mathrm{Var}[W_n] = O(1)$, $\mathbb{E}[W_n^3] = O(1)$. Then, applying the Berry-Esseen theorem, we get:

$$p_1 = \Pr\left[S_k < k \cdot \sigma_Q^2\right] = \Phi(0) + O\left(\frac{1}{\sqrt{k}}\right) = \frac{1}{2} + O\left(\frac{1}{\sqrt{k}}\right) \,,$$

$$p_2 = \Pr\left[T_k < k \cdot \sigma_Q^2\right] = \Phi\left(\frac{k \cdot \sigma_Q^2 - \mathbb{E}[T_k]}{\sqrt{\mathrm{Var}[T_k]}}\right) + O\left(\frac{1}{\sqrt{k}}\right) = \Phi\left(O\left(\frac{1}{\sqrt{k}}\right)\right) + O\left(\frac{1}{\sqrt{k}}\right)$$

$$= \frac{1}{2} + O\left(\frac{1}{\sqrt{k}}\right) \,.$$

And thus,

$$\Delta_{\mathrm{TV}}(\mathcal{P}, \mathcal{Q}) = p_1 - p_2 = O(1/\sqrt{k}) \,. \tag{5}$$

$\square$

We now define a rotation-translation adversary $\mathcal{A}$ with a budget of $N$. It samples a random permutation from the set $\Pi$ of permutations that switch position 0 with a position in $[0, N-1]$ and leave all other positions fixed (note that $|\Pi| = N$). Let $\mathcal{A}(\mathcal{G}^y, \mathcal{H}^y)$ denote the distribution resulting from applying $\mathcal{A}$ to $(\mathcal{G}^y, \mathcal{H}^y)$ and define $\mathcal{A}(\mathcal{G}^{-y}, \mathcal{H}^y)$ similarly. Recall that $\mathcal{Z}^y$ is a hybrid distribution which has all features distributed as $\mathcal{N}(y\eta, 1)$.

**Claim 7.** $\Delta_{TV}\left(\mathcal{A}(\mathcal{G}^y, \mathcal{H}^y), \mathcal{Z}^y\right) = O(1/\sqrt{N})$ and $\Delta_{TV}\left(\mathcal{A}(\mathcal{G}^{-y}, \mathcal{H}^y), \mathcal{Z}^y\right) = O(1/\sqrt{N})$

*Proof.* For the first $N$ features, samples output by $\mathcal{A}$ follow exactly the distribution $\mathcal{Q}$ from Lemma (6), for $k = N$ and $\lambda_P = y \cdot \eta, \lambda_Q = y, \sigma_Q^2 = \alpha^{-2}$. Note that in this case, the distribution $\mathcal{P}$ has each feature distributed as in $\mathcal{Z}^y$. Thus, Lemma (6) tells us that the distribution of the first $N$ features is the same as in $\mathcal{Z}^y$, up to a total-variation distance of $O(1/\sqrt{N})$. As features $x_N \ldots, x_d$ are unaffected by $\mathcal{A}$ and thus remain distributed as in $\mathcal{Z}^y$, we conclude that the total-variation distance between $\mathcal{A}$'s outputs and $\mathcal{Z}^y$ is $O(1/\sqrt{N})$.

The proof for $\mathcal{A}(\mathcal{G}^{-y}, \mathcal{H}^y)$ is similar, except that we apply Lemma (6) with $\lambda_Q = -y$. $\square$

Let $\tilde{\mathcal{Z}}^y$ be the true distribution $\mathcal{A}(\mathcal{G}^{-y}, \mathcal{H}^y)$, which we have shown to be close to $\mathcal{Z}^y$. Consider the following "inverse" adversary $\mathcal{A}^{-1}$. This adversary samples $\boldsymbol{z} \sim \tilde{\mathcal{Z}}^y$ and returns $\pi^{-1}(\boldsymbol{z})$, for $\pi \in \Pi$, with probability

$$\frac{1}{|\Pi|} \cdot \frac{f_{(\mathcal{G}^{-y}, \mathcal{H}^y)}(\pi^{-1}(\boldsymbol{z}))}{f_{\tilde{\mathcal{Z}}^y}(\boldsymbol{z})} \,,$$

where $f_{(\mathcal{G}^{-y}, \mathcal{H}^y)}$ and $f_{\tilde{\mathcal{Z}}^y}$ are the probability density functions for $(\mathcal{G}^{-y}, \mathcal{H}^y)$ and for $\tilde{\mathcal{Z}}^y$.

**Claim 8.** $\mathcal{A}^{-1}$ *is a RT adversary with budget* $N$ *that transforms* $\tilde{\mathcal{Z}}^y$ *into* $(\mathcal{G}^{-y}, \mathcal{H}^y)$.

571 *Proof.* Note that $\mathcal{A}^{-1}$ always applies the inverse of a perturbation in $\Pi$. So feature $x_0$ gets sent to at
572 most $N$ positions when perturbed by $\mathcal{A}^{-1}$.

573 Let $Z$ be a random variable distributed as $\tilde{\mathcal{Z}}^y$ and let $h$ be the density function of the distribution
574 obtained by applying $\mathcal{A}^{-1}$ to $Z$. We compute:

$$h(\boldsymbol{x}) = \sum_{\pi \in \Pi} f_{\tilde{\mathcal{Z}}^y}(\pi(\boldsymbol{x})) \cdot \Pr\left[\mathcal{A}^{-1} \text{ picks permutation } \pi \mid Z = \pi(\boldsymbol{x})\right]$$

$$= \sum_{\pi \in \Pi} f_{\tilde{\mathcal{Z}}^y}(\pi(\boldsymbol{x})) \cdot \frac{1}{|\Pi|} \cdot \frac{f_{(\mathcal{G}^{-y}, \mathcal{H}^y)}(\pi(\pi^{-1}(\boldsymbol{x})))}{f_{\tilde{\mathcal{Z}}^y}(\pi(\boldsymbol{x}))} = \sum_{\pi \in \Pi} \frac{1}{|\Pi|} \cdot f_{(\mathcal{G}^{-y}, \mathcal{H}^y)}(\boldsymbol{x})$$

$$= f_{(\mathcal{G}^{-y}, \mathcal{H}^y)}(\boldsymbol{x}) \,,$$

575 so applying $\mathcal{A}^{-1}$ to $\tilde{\mathcal{Z}}^y$ does yield the distribution $(\mathcal{G}^{-y}, \mathcal{H}^y)$. $\square$

576 We can now finally define our main rotation-translation adversary, $\mathcal{A}^*$. The adversary first applies $\mathcal{A}$
577 to samples from $(\mathcal{G}^y, \mathcal{H}^y)$, and then applies $\mathcal{A}^{-1}$ to the resulting samples from $\tilde{\mathcal{Z}}^y$.

578 **Claim 9.** *The adversary $\mathcal{A}^*$ is a rotation-translation adversary with budget $N$. Moreover,*
579 $\Delta_{TV}\left(\mathcal{A}^*(\mathcal{G}^y, \mathcal{H}^y), (\mathcal{G}^{-y}, \mathcal{H}^y)\right) = O(1/\sqrt{N})$.

580 *Proof.* The adversary $\mathcal{A}^*$ first switches $x_0$ with some random position in $[0, N-1]$ by applying $\mathcal{A}$.
581 Then, $\mathcal{A}^{-1}$ either switches $x_0$ back into its original position or leaves it untouched. Thus, $\mathcal{A}^*$ always
582 moves $x_0$ into one of $N$ positions. The total-variation bound follows by the triangular inequality:

$$\Delta_{\mathrm{TV}}\left(\mathcal{A}^*(\mathcal{G}^y, \mathcal{H}^y), (\mathcal{G}^{-y}, \mathcal{H}^y)\right)$$

$$= \Delta_{\mathrm{TV}}\left(\mathcal{A}^{-1}(\mathcal{A}(\mathcal{G}^y, \mathcal{H}^y)), (\mathcal{G}^{-y}, \mathcal{H}^y)\right)$$

$$\leq \Delta_{\mathrm{TV}}\left(\mathcal{A}^{-1}(\mathcal{Z}^y), (\mathcal{G}^{-y}, \mathcal{H}^y)\right) + \Delta_{\mathrm{TV}}\left(\mathcal{Z}^y, \mathcal{A}(\mathcal{G}^y, \mathcal{H}^y)\right)$$

$$\leq \underbrace{\Delta_{\mathrm{TV}}\left(\mathcal{A}^{-1}(\tilde{\mathcal{Z}}^y), (\mathcal{G}^{-y}, \mathcal{H}^y)\right)}_{0} + \underbrace{\Delta_{\mathrm{TV}}\left(\tilde{\mathcal{Z}}^y, (\mathcal{G}^{-y}, \mathcal{H}^y)\right)}_{O(1/\sqrt{N})} + \underbrace{\Delta_{\mathrm{TV}}\left(\mathcal{Z}^y, \mathcal{A}(\mathcal{G}^y, \mathcal{H}^y)\right)}_{O(1/\sqrt{N})}$$

$$= O(1/\sqrt{N}) \,.$$

583 $\square$

584 To conclude the proof, we define:

$$p_{+-} = \Pr_{\boldsymbol{x} \sim (\mathcal{G}^{+1}, \mathcal{H}^{-1})}[f(\boldsymbol{x}) = +1] \,, \qquad p_{-+} = \Pr_{\boldsymbol{x} \sim (\mathcal{G}^{-1}, \mathcal{H}^{+1})}[f(\boldsymbol{x}) = +1] \,,$$

$$\tilde{p}_{-+} = \Pr_{\boldsymbol{x} \sim \mathcal{A}^*(\mathcal{G}^{+1}, \mathcal{H}^{+1})}[f(\boldsymbol{x}) = +1] \,, \qquad \tilde{p}_{+-} = \Pr_{\boldsymbol{x} \sim (\mathcal{G}^{-1}, \mathcal{H}^{-1})}[f(\boldsymbol{x}) = +1] \,.$$

585 Then,

$$\Pr[f(\boldsymbol{x} + \boldsymbol{r}_\infty) = y] + \Pr[f(A^*(\boldsymbol{x})) = y] = \frac{1}{2}p_{+-} + \frac{1}{2}(1 - p_{-+}) + \frac{1}{2}\tilde{p}_{-+} + \frac{1}{2}(1 - \tilde{p}_{+-})$$

$$= 1 + \frac{1}{2}(p_{+-} - \tilde{p}_{+-}) + \frac{1}{2}(p_{-+} - \tilde{p}_{-+})$$

$$\leq 1 - O(1/\sqrt{N}) \,.$$

586 $\square$

## G.1 Numerical Estimates for the Robustness Trade-off in Theorem 2

588 While the robustness trade-off we proved in Theorem 2 is asymptotic in $N$ (the budget of the RT
589 adversary), we can provide tight numerical estimates for this trade-off for concrete parameter settings:

590 **Remark 10.** Let $d \geq 200$, $\alpha = 2$ and $N = 49$ (e.g., translations by $\pm 3$ pixels). Then, there exists a
591 classifier with $\mathcal{R}_{\mathrm{adv}}(f; S_\infty) < 10\%$, as well as a (distinct) classifier with $\mathcal{R}_{\mathrm{adv}}(f; S_{\mathrm{RT}}) < 10\%$. Yet,
592 any single classifier satisfies $\mathcal{R}_{\mathrm{adv}}^{\mathrm{avg}}(f; S_\infty, S_{\mathrm{RT}}) \gtrapprox 0.425$.

593 We first show the existence of classifiers with $\mathcal{R}_{\text{adv}} < 10\%$ for the given $\ell_\infty$ and RT attacks.

594 Let $f(\boldsymbol{x}) = \text{sign}(x_0)$ and let $\boldsymbol{r} = [-y\epsilon, 0, \ldots, 0]$ be the worst-case perturbation with $\|\boldsymbol{r}\| \leq \epsilon$. Recall
595 that $\epsilon = 2\eta = 4/\sqrt{d}$. We have

$$\Pr[f(\boldsymbol{x} + \boldsymbol{r}) \neq y] = \Pr\left[\mathcal{N}(1, 1/4) - 4/\sqrt{d} < 0\right] \leq \Pr\left[\mathcal{N}(1 - 4/\sqrt{200}, 1/4) < 0\right] \leq 8\% \;.$$

596 Thus, $f$ achieves $\mathcal{R}_{\text{adv}} < 10\%$ against the $\ell_\infty$-perturbations.

597 Let $g(\boldsymbol{x}) = \text{sign}(\sum_{i=N}^{d} x_i)$ be a classifier that ignores all feature positions that a RT adversary $\mathcal{A}$
598 may affect. We have

$$\Pr[g(\mathcal{A}(\boldsymbol{x})) \neq y] = \Pr[g(\boldsymbol{x}) \neq y] = \Pr\left[\mathcal{N}\left((d - N + 1) \cdot \eta, d - N + 1\right) < 0\right]$$
$$\leq \Pr\left[\mathcal{N}(2\sqrt{d - 48}/\sqrt{d}, 1) < 0\right] \leq 5\% \;.$$

599 Thus, $g$ achieves $\mathcal{R}_{\text{adv}} < 10\%$ against RT perturbations.

600 We upper-bound the adversarial risk that any classifier must incur against both attacks by numerically
601 estimating the total-variation distance between the distributions induced by the RT and $\ell_\infty$ adversaries
602 for inputs of opposing labels $y$. Specifically, we generate 100,000 samples from the distributions
603 $\mathcal{G}^{+1}, \mathcal{G}^{-1}$ and $\mathcal{H}^{+1}$ as defined in the proof of Theorem 2, and obtain an estimate of the total-variation
604 distance in Lemma (9). For this, we numerically estimate $p_1$ and $p_2$ as defined in Equation (4).

## H Proof of Claim 3 (Affine combinations of $\ell_p$- perturbations do not affect linear models)

607 Let

$$\max_{\boldsymbol{r} \in S_U} \boldsymbol{w}^T \boldsymbol{r} = v_{\max}, \quad \text{and} \quad \min_{\boldsymbol{r} \in S_U} \boldsymbol{w}^T \boldsymbol{r} = v_{\min} \;.$$

608 Let $S_U := S_p \cup S_q$. Note that any $\boldsymbol{r} \in S_{\text{affine}}$ is of the form $\beta \boldsymbol{r}_1 + (1 - \beta) \boldsymbol{r}_2$ for $\beta \in [0, 1]$. Moreover,
609 we have $\boldsymbol{r}_1 \in S_p \subset S_U$ and $\boldsymbol{r}_2 \in S_q \subset S_U$. Thus,

$$\max_{\boldsymbol{r} \in S_{\text{affine}}} \boldsymbol{w}^T \boldsymbol{r} = v_{\max}, \quad \text{and} \quad \min_{\boldsymbol{r} \in S_{\text{affine}}} \boldsymbol{w}^T \boldsymbol{r} = v_{\min} \;.$$

610 Let $h(\boldsymbol{x}) = \boldsymbol{w}^T \boldsymbol{x} + b$, so that $f(\boldsymbol{x}) = \text{sign}(h(\boldsymbol{x}))$. Then, we get

$$\Pr_{\mathcal{D}}\left[\exists \boldsymbol{r} \in S_{\text{affine}} : f(\boldsymbol{x} + \boldsymbol{r}) \neq y\right] = \frac{1}{2} \Pr_{\mathcal{D}}\left[\exists \boldsymbol{r} \in S_{\text{affine}} : \boldsymbol{w}^T \boldsymbol{r} < -h(\boldsymbol{x}) \mid y = +1\right]$$
$$+ \frac{1}{2} \Pr_{\mathcal{D}}\left[\exists \boldsymbol{r} \in S_{\text{affine}} : \boldsymbol{w}^T \boldsymbol{r} > h(\boldsymbol{x}) \mid y = -1\right]$$
$$= \frac{1}{2} \Pr_{\mathcal{D}}\left[v_{\min} < -h(\boldsymbol{x}) \mid y = +1\right] + \frac{1}{2} \Pr_{\mathcal{D}}\left[v_{\max} > h(\boldsymbol{x}) \mid y = -1\right]$$
$$= \frac{1}{2} \Pr_{\mathcal{D}}\left[\exists \boldsymbol{r} \in S_U : \boldsymbol{w}^T \boldsymbol{r} < -h(\boldsymbol{x}) \mid y = +1\right]$$
$$+ \frac{1}{2} \Pr_{\mathcal{D}}\left[\exists \boldsymbol{r} \in S_U : \boldsymbol{w}^T \boldsymbol{r} > h(\boldsymbol{x}) \mid y = -1\right]$$
$$= \Pr_{\mathcal{D}}\left[\exists \boldsymbol{r} \in S_U : f(\boldsymbol{x} + \boldsymbol{r}) \neq y\right] \;.$$

611 $\qquad\qquad\qquad\qquad\qquad\qquad\qquad\qquad\qquad\qquad\qquad\qquad\qquad\qquad\qquad\qquad\qquad\qquad$ $\square$

## I Affine combinations of $\ell_p$- perturbations can affect non-linear models

613 In Section 2.5, we showed that for linear models, robustness to a union of $\ell_p$-perturbations implies
614 robustness to an affine adversary that interpolates between perturbation types. We show that this need
615 not be the case when the model is non-linear. In particular, we can show that for the distribution
616 $\mathcal{D}$ introduced in Section 2, non-linearity is necessary to achieve robustness to a union of $\ell_\infty$ and
617 $\ell_1$-perturbations (with different parameter settings than for Theorem 1), but that at the same time,
618 robustness to affine combinations of these perturbations is unattainable by any model.

**Theorem 11.** *Consider the distribution $\mathcal{D}$ with $d \geq 200$, $\alpha = 2$ and $p_0 = 1 - \Phi(-2)$. Let $S_\infty$ be the set of $\ell_\infty$-bounded perturbation with $\epsilon = (3/2)\eta = 3/\sqrt{d}$ and let $S_1$ be the set of $\ell_1$-bounded perturbations with $\epsilon = 3$. Define $S_{affine}$ as in Section 2.5. Then, there exists a non-linear classifier $g$ that achieves $\mathcal{R}_{adv}^{max}(g; S_\infty, S_1) \leq 35\%$. Yet, for all classifiers $f$ we have $\mathcal{R}_{adv}(f; S_{affine}) \geq 50\%$.*

*Proof.* We first prove that no classifier can achieve accuracy above $50\%$ (which is achieved by the constant classifier) against $S_{\text{affine}}$. The proof is very similar to the one of Theorem 1.

Let $\beta = 2/3$, so the affine attacker gets to compose an $\ell_\infty$-budget of $2/\sqrt{d}$ and an $\ell_1$-budget of $1$. Specifically, for a point $(\boldsymbol{x}, y) \sim \mathcal{D}$, the affine adversary will apply the perturbation

$$\boldsymbol{r} = [-x_0, -y\frac{2}{\sqrt{d}}, \ldots, -y\frac{2}{\sqrt{d}}] = [-x_0, -y\eta, \ldots, -y\eta] \ .$$

Let $\mathcal{G}^{0,0}$ be the following distribution:

$$y \stackrel{u.a.r}{\sim} \{-1, +1\}, \quad x_0 = 0, \quad x_1, \ldots, x_d \stackrel{i.i.d}{\sim} \mathcal{N}(0,1) \ .$$

Note that in $\mathcal{G}^{0,0}$, $\boldsymbol{x}$ is independent of $y$ so no classifier can achieve more than $50\%$ accuracy on $\mathcal{G}^{0,0}$. Yet, note that the affine adversary's perturbation $\boldsymbol{r}$ transforms any $(\boldsymbol{x}, y) \sim \mathcal{D}$ into $(\boldsymbol{x}, y) \sim \mathcal{G}^{0,0}$.

We now show that there exists a classifier that achieves non-trivial robustness against the set of perturbations $S_\infty \cup S_1$, i.e., the union of $\ell_\infty$-noise with $\epsilon = 3/\sqrt{d}$ and $\ell_1$-noise with $\epsilon = 3$. Note that by Claim 3, this classifier must be *non-linear*. We define

$$f(\boldsymbol{x}) = \text{sign}\left(3 \cdot \text{sign}(x_0) + \sum_{i=1}^{d} \frac{2}{\sqrt{d}} \cdot x_i\right) \ .$$

The reader might notice that $f(\boldsymbol{x})$ closely resembles the *Bayes optimal classifier* for $\mathcal{D}$ (which would be a linear classifier). The non-linearity in $f$ comes from the sign function applied to $x_0$. Intuitively, this limits the damage caused by the $\ell_1$-noise, as $\text{sign}(x_0)$ cannot change by more than $\pm 2$ under any perturbation of $x_0$. This forces the $\ell_1$ perturbation budget to be "wasted" on the other features $x_1, \ldots, x_d$, which are very robust to $\ell_1$ attacks.

As a warm-up, we compute the classifier's natural accuracy on $\mathcal{D}$. For $(\boldsymbol{x}, y) \sim \mathcal{D}$, let $X = y \cdot \sum_{i=1}^{d} \frac{2}{\sqrt{d}} \cdot x_i$ be a random variable. Recall that $\eta = 2/\sqrt{d}$. Note that $X$ is distributed as

$$y \cdot \sum_{i=1}^{d} \frac{2}{\sqrt{d}} \cdot \mathcal{N}(y\eta, 1) = \sum_{i=1}^{d} \frac{2}{\sqrt{d}} \cdot \mathcal{N}\left(\frac{2}{\sqrt{d}}, 1\right) = \sum_{i=1}^{d} \mathcal{N}\left(\frac{4}{d}, \frac{4}{d}\right) = \mathcal{N}(4, 4) \ .$$

Recall that $x_0 = y$ with probability $p_0 = 1 - \Phi(-2) \approx 0.977$. We get:

$$
\begin{aligned}
\Pr_{\mathcal{D}}[f(\boldsymbol{x}) = y] &= \Pr_{\mathcal{D}}\left[y \cdot \left(3 \cdot \text{sign}(x_0) + \sum_{i=1}^{d} \frac{2}{\sqrt{d}} \cdot x_i\right) > 0\right] \\
&= \Pr_{\mathcal{D}}[x_0 = y] \cdot \Pr_{\mathcal{D}}[3 \cdot y \cdot \text{sign}(x_0) + X > 0 \mid x_0 = y] \\
&\quad + \Pr_{\mathcal{D}}[x_0 \neq y] \cdot \Pr_{\mathcal{D}}[3 \cdot y \cdot \text{sign}(x_0) + X > 0 \mid x_0 \neq y] \\
&= p \cdot \Pr[3 + \mathcal{N}(4, 4) > 0] + (1 - p) \cdot \Pr[-3 + \mathcal{N}(4, 4) > 0] \approx 99\% \ .
\end{aligned}
$$

We now consider an adversary that picks either an $\ell_\infty$-perturbation with $\epsilon = 3/\sqrt{d}$ or an $\ell_1$-perturbation with $\epsilon = 3$. It will suffice to consider the case where $x_0 = y$. Note that the $\ell_\infty$ classifier cannot meaningfully perturb $x_0$, and the best perturbation is always $\boldsymbol{r}_\infty = [0, -y3/\sqrt{d}, \ldots, -y3/\sqrt{d}]$. Moreover, the best $\ell_1$-bounded perturbation is $\boldsymbol{r}_1 = [-2y, -y, 0, \ldots, 0]$. We have $f(\boldsymbol{x} + \boldsymbol{r}_\infty) = \text{sign}(y \cdot (3 + X - 6))$ and $f(\boldsymbol{x} + \boldsymbol{r}_1) = \text{sign}(y \cdot (-3 + X - 2/\sqrt{d}))$. We now lower-bound the classifier's accuracy under the union $S_{\text{U}} := S_\infty \cup S_1$ of these two perturbation models:

$$
\begin{aligned}
\Pr_{\mathcal{D}}[f(\boldsymbol{x} + \boldsymbol{r}) = y, \forall \boldsymbol{r} \in S_{\text{U}}] &\geq \Pr_{\mathcal{D}}[x_0 = y] \cdot \Pr_{\mathcal{D}}[f(\boldsymbol{x} + \boldsymbol{r}) = y, \forall \boldsymbol{r} \in S_{\text{U}} \mid x_0 = y] \\
&\geq p \cdot \Pr_{\mathcal{D}}\left[(3 + X - 6 > 0) \wedge (-3 + X - 2/\sqrt{d}) > 0)\right] \\
&= p \cdot \Pr\left[\mathcal{N}(4, 4) > 3 + 2/\sqrt{d}\right] \geq 65\% \quad (\text{for } d \geq 200) \ .
\end{aligned}
$$

$\square$

## J Proof of Theorem 4 (Affine combinations of $\ell_\infty$- and spatial perturbations can affect linear models)

Note that our definition of affine perturbation allows for a different weighting parameter $\beta$ to be chosen for each input. Thus, the adversary that selects perturbations from $S_{\text{affine}}$ is at least as powerful as the one that selects perturbations from $S_\infty \cup S_{\text{RT}}$. All we need to show to complete the proof is that there exists some input $x$ that the affine adversary can perturb, while the adversary limited to the union of spatial and $\ell_\infty$ perturbations cannot.

Without loss of generality, assume that the RT adversary picks a permutation that switches $x_0$ with a position in $[0, N-1]$, and leaves all other indices untouched. The main idea is that for any input $x$ where the RT adversary moves $x_0$ to position $j < N-1$, the RT adversary with budget $N$ is no more powerful than one with budget $j+1$. The affine adversary can thus limit its rotation-translation budget and use the remaining budget on an extra $\ell_\infty$ perturbation.

We now construct an input $x$ such that: (1) $x$ cannot be successfully attacked by an RT adversary (with budget $N$) or by an $\ell_\infty$-adversary (with budget $\epsilon$); (2) $x$ can be attacked by an affine adversary.

Without loss of generality, assume that $w_1 = \min\{w_1, \ldots, w_{N-1}\}$, i.e., among all the features that $x_0$ can be switched with, $x_1$ has the smallest weight. Let $y = +1$, and let $x_1, \ldots, x_{N-1}$ be chosen such that $\arg\min\{x_1, \ldots, x_{N-1}\} = 1$. We set

$$x_0 := \frac{\epsilon \cdot \|\boldsymbol{w}\|_1}{w_0 - w_1} + x_1 .$$

Moreover, set $x_N, \ldots, x_d$ such that

$$\boldsymbol{w}^T \boldsymbol{x} + b = 1.1 \cdot \epsilon \cdot \|\boldsymbol{w}\|_1 .$$

Note that constructing such an $x$ is always possible as we assumed $w_0 > w_i > 0$ for all $1 \le i \le d$.

We now have an input $(x, y)$ that has non-zero support under $\mathcal{D}$. Let $r$ be a perturbation with $\|\boldsymbol{r}\|_\infty \le \epsilon$. We have:

$$\boldsymbol{w}^T(\boldsymbol{x} + \boldsymbol{r}) + b \ge \boldsymbol{w}^T \boldsymbol{x} + b - \epsilon \cdot \|\boldsymbol{w}\|_1 = 0.1 \cdot \epsilon \cdot \|\boldsymbol{w}\|_1 > 0 ,$$

so $f(\boldsymbol{w}^T(\boldsymbol{x} + \boldsymbol{r}) + b) = y$, i.e., $x$ cannot be attacked by any $\epsilon$-bounded $\ell_\infty$-perturbation.

Define $\hat{\boldsymbol{x}}_i$ as the input $x$ with features $x_0$ and $x_i$ switched, for some $0 \le i < N$. Then,

$$\begin{aligned}
\boldsymbol{w}^T \hat{\boldsymbol{x}}_i + b &= \boldsymbol{w}^T \boldsymbol{x} + b - (w_0 - w_i) \cdot (x_0 - x_i) \\
&\ge \boldsymbol{w}^T \boldsymbol{x} + b - (w_0 - w_1) \cdot (x_0 - x_1) \\
&= \boldsymbol{w}^T \boldsymbol{x} + b - \epsilon \cdot \|\boldsymbol{w}\|_1 = 0.1 \cdot \epsilon \cdot \|\boldsymbol{w}\|_1 > 0 .
\end{aligned}$$

Thus, the RT adversary cannot change the sign of $f(\boldsymbol{x})$ either. This means that an adversary that chooses from $S_\infty \cup S_{\text{RT}}$ cannot successfully perturb $x$.

Now, consider the affine adversary, with $\beta = 2/N$ that first applies an RT perturbation with budget $\frac{2}{N} \cdot N = 2$ (i.e., the adversary can only flip $x_0$ with $x_1$), followed by an $\ell_\infty$-perturbation with budget $(1 - \frac{2}{N}) \cdot \epsilon$. Specifically, the adversary flips $x_0$ and $x_1$ and then adds noise $\boldsymbol{r} = -(1 - \frac{2}{N}) \cdot \epsilon \cdot \text{sign}(\boldsymbol{w})$. Let this adversarial example by $\hat{\boldsymbol{x}}_{\text{affine}}$. We have

$$\begin{aligned}
\boldsymbol{w}^T \hat{\boldsymbol{x}}_{\text{affine}} + b &= \boldsymbol{w}^T \boldsymbol{x} + b - (w_0 - w_1) \cdot (x_0 - x_1) - \left(1 - \frac{2}{N}\right) \cdot \epsilon \cdot \|\boldsymbol{w}\|_1 \\
&= 1.1 \cdot \epsilon \cdot \|\boldsymbol{w}\|_1 - \epsilon \cdot \|\boldsymbol{w}\|_1 - \left(1 - \frac{2}{N}\right) \cdot \epsilon \cdot \|\boldsymbol{w}\|_1 \\
&= -\left(0.9 - \frac{2}{N}\right) \cdot \epsilon \cdot \|\boldsymbol{w}\|_1 \\
&< 0 .
\end{aligned}$$

Thus, $f(\hat{\boldsymbol{x}}_{\text{affine}}) = -1 \ne y$, so the affine adversary is strictly stronger that the adversary that is restricted to RT or $\ell_\infty$ perturbations. $\square$

## Footnotes

[4]Our new attack $\ell_1$-attack, described in Section 3, has a parameter $q$ to controls the sparsity of the gradient updates. When leaving this parameter constant during training, the model overfits and fails to achieve general robustness. To resolve this issue, we sample $q \in [80\%, 99.5\%]$ at random for each attack during training. We also found that 10 iterations were insufficient to get a strong attack and thus increased the iteration count to 20.

[5]Specifically, for the "secret" model of Madry et al., the three thresholding filters are approximately $\text{ReLU}(0.6 \cdot (x - 0.3))$, $\text{ReLU}(1.34 \cdot (x - 0.3))$ and $\text{ReLU}(0.86 \cdot (x - 0.7))$.

[6] We thank Iosif Pinelis for his help with this proof (`https://mathoverflow.net/questions/325409/`).