[Reviews · NeurIPS 2019]

Reviewer 1



The theoretical contribution of this paper (Section 2) is solid and neatly extends the work of Tsipras et al. (ICLR, 2019) to simultaneous robustness to multiple perturbations. The empirical work about comparison of different attacks and simultaneous robustness of models to them is also interesting, and raises important questions (e.g., multi-perturbation robust models for MNIST and CIFAR 10). The presentation is clear, and I personally liked the problem and their results.

Reviewer 2



Originality: the paper is mostly original in considering the problem of robustness to multiple perturbation types. The trade-off between adversarial robustness to different norms and the definition of "affine attacks" has been also investigated for linear classifiers in: - A. Demontis, P. Russu, B. Biggio, G. Fumera, and F. Roli. On security and sparsity of linear classifiers for adversarial settings. In A. Robles-Kelly, M. Loog, B. Biggio, F. Escolano, and R. Wilson, editors, Joint IAPR Int’l Workshop on Structural, Syntactic, and Statistical Pattern Recognition, volume 10029 of LNCS, pages 322–332, Cham, 2016. Springer International Publishing. In that paper, it is shown that while one can design an optimal classifier against one lp-norm attack, the same classifier will be vulnerable to the corresponding dual-norm attack (e.g., if one designs a robust classifier against l-inf attacks, it will be vulnerable to l1 attacks). This result was based on the paper: - H. Xu, C. Caramanis, and S. Mannor. Robustness and regularization of support vector machines. Journal of Machine Learning Research, 10:1485–1510, July 2009. where the authors established an equivalence between 'adversarial training' (i.e., solving a non-regularized, robust optimization problem) and a regularized linear SVM. In other words, it is shown that a proper regularizer can be the optimal response to a specific lp-norm attack. In the submitted paper, this phenomenon is stated in terms of mutually exclusive perturbations (MEPs) and shown for a toy Gaussian dataset. I think it would be very interesting to explore connections to the notion of dual norms as explored in the aforementioned papers. This at least deserves to be discussed in the paper. Quality: the submission is technically sound, even though I did not check all the derivations in detail. Clarity: clarity could be improved. The paper provides mixed heterogeneous contributions which are somewhat listed in the paper in a not completely structured manner. It would have been useful to try to better separate the main contributions and the corresponding experimental analyses. I also understand however that this may be problematic, as space is quite limited and the contributions of this paper are somewhat heterogeneous in nature. Significance: the results are interesting for the community working on adversarial examples. Even though the notion of trade-off between different attacks has been around for a while, it has not been clearly analyzed as done in this paper. COMMENTS AFTER READING THE AUTHORS' REBUTTAL -------------- I read the authors' response and thank them for welcoming my suggestions. Another suggestion that they may find useful is to read a recent survey which clarifies that adversarial machine learning started much earlier than 2014, and gradient-based adversarial examples were essentially re-discovered in 2014. - B. Biggio and F. Roli. Wild patterns: Ten years after the rise of adversarial machine learning. Pattern Recognition, 84:317–331, 2018. In fact, similar gradient-based attacks were already developed by Biggio et al., who were actually the first to formulate adversarial attacks as optimization problems and solve them via gradient-based optimization (in particular, see: - B. Biggio et al., Evasion attacks against ML at test time, ECML PKDD 2013 - B. Biggio et al., Poisoning attacks against SVMs in ICML 2012 This is also mentioned by Ian Goodfellow in some of his talks (see, e.g., his talk at Stanford - https://www.youtube.com/watch?v=CIfsB_EYsVI) and acknowledged in the book: - A. D. Joseph, B. Nelson, B. I. P. Rubinstein, and J. Tygar. Adversarial Machine Learning. Cambridge University Press, 2018.

Reviewer 3



The paper follows the work of Tsipras et. al., and use simple data distributions to theoretically study the robustness trade-offs. The hypothesis is that adversarially-trained models tend to focus on different features to achieve robustness to a particular attack. The paper shows, through analysis of the activations of the network, that robustness to l_\infty leads to gradient masking for other types of adversaries on MNIST. Looking at Table 5., this does not seem to be the case for CIFAR. Furthermore, the trade-off seems to be less significant on this dataset (e.g., comparing OPT(R^{avg}) and R^{avg}). This suggests that some of the results may be artifacts of MNIST, and more generally, the hypothesis in Tsipras et. al. on explaining the phenomenon of adversarial examples may not be sufficient.

[Author Response · NeurIPS 2019]

We thank all reviewers for their positive reception of our paper and for their constructive feedback.

**Reviewer #2**

- **On dual norms and prior work.** Thank you for pointing us to the relevant prior work of Demontis et al. and Xu et al. which we apparently missed.

  Our results about the tradeoff between $\ell_1$ and $\ell_\infty$ robustness do indeed share similarities with the dual-norm behavior studied by Demontis et al. for linear classifiers. The main difference is that for our Gaussian dataset, we show that the robustness tradeoff is inherent to *any* classifier (i.e., not only linear ones). Another difference is that we extend our results to tradeoffs between $\ell_\infty$ and spatial perturbations.

  For arbitrary dual norms $\ell_p$ and $\ell_q$, we can also exhibit a tradeoff as follows (somewhat informally):

  - Flipping the features $x_1, \ldots, x_n$ requires a $\ell_p$-perturbation of magnitude $O(\mu \cdot n^{1/p}) = O(n^{1/p-1/2})$.
  - Flipping the feature $x_0$ requires a $\ell_q$-perturbation of magnitude $O(1)$.
  - So if a model is robust to perturbations of size $O(n^{1/p-1/2})$ in the $\ell_p$-norm, it cannot also be robust to perturbations of size $O(1)$ in the dual norm.

  We will discuss these connections between our work and the prior work of Demontis et al. and Xu et al. in the final version of our paper. We thank the reviewer for suggesting this dual-norm view which nicely generalizes one of our results.

- **On structure and readability.** We agree that our paper contains many contributions (the formal analysis, a new $\ell_1$-attack, an experimental evaluation) that are somewhat heterogenous. We will make an effort to clarify our main contributions and to better structure our paper to improve its readability.

**Reviewer #3**

- **On MNIST artifacts.** The gradient masking effect we discover and explain is indeed specific to MNIST (for multiple $\ell_p$ norms), and we do not claim otherwise. In fact, this gradient masking effect seems inherently due to the mostly binary nature of the MNIST images, which leads to thresholding being a viable defense against $\ell_\infty$-perturbations.

  Nevertheless, as MNIST is the only vision dataset for which we've been able to train models to high levels of robustness (for individual attack models), we believe it is worthwhile to observe that extending this robustness to multiple $\ell_p$ attacks may be particularly challenging for this dataset. In this sense, even a dataset as simple as MNIST is clearly not solved from an adversarial robustness perspective.

  We do believe that it should be possible to train MNIST models to a robustness tradeoff similar to that we found on CIFAR10. But this will require new techniques that somehow circumvent gradient masking as a spurious solution. We think this is an interesting open problem for the community to consider.

  On CIFAR10, the robustness tradeoff is indeed smaller but still quite noticeable. It is worth noting that our experiments on CIFAR10 only ever consider the combination of two attack types. It is not clear what would happen if we were to try to train models to be robust to $4$ or $5$ attacks at a time for instance. For these types of experiments we are mainly limited by the poor scalability of adversarial training (e.g., for two attack types, adversarially training a wide ResNet takes about two GPU weeks). There is some promising recent research on speeding up adversarial training, so these types of experiments might become tractable for future work.

- **On black-box attacks.** The Ensemble Adversarial Training technique of Tramer et al. was proposed to increase a model's robustness to black-box attacks, but it was found to have no noticeable effect on the model's robustness to stronger white-box attacks. As our evaluation focuses on the white-box robustness of the trained models, we have not incorporated black-box attack examples at training time.

  We also considered using black-box attacks at evaluation time (e.g., as a test against gradient-masking), but found decision-based attacks to be stronger and more reliable for this purpose.

[Meta-Review · NeurIPS 2019]

Thank you for your submission to NeurIPS. All reviewers and I are positive about this paper, and I am happy to recommend that it be accepted. One reviewer in particular was more positive about the paper after the discussion presented in the author response, so the one comment would be to include these discussion points in the paper itself.